# Stabilizing lattice oxygen redox in layered sodium transition metal oxide through spin singlet state

Xuelong Wang[1,2], Liang Yin[2,3], Arthur Ronne[1,4], Yiman Zhang[5], Zilin Hu[2], Sha Tan[1], Qinchao Wang[1], Bohang Song[6], Mengya Li[7], Xiaohui Rong[2], Saul Lapidus[3], Shize Yang[8], Enyuan Hu[1] ✉ & Jue Liu[6] ✉

Reversible lattice oxygen redox reactions offer the potential to enhance energy density and lower battery cathode costs. However, their widespread adoption faces obstacles like substantial voltage hysteresis and poor stability. The current research addresses these challenges by achieving a non-hysteresis, long-term stable oxygen redox reaction in the P3-type $Na_{2/3}Cu_{1/3}Mn_{2/3}O_2$. Here we show this is accomplished by forming spin singlet states during charge and discharge. Detailed analysis, including in-situ X-ray diffraction, shows highly reversible structural changes during cycling. In addition, local $CuO_6$ Jahn-Teller distortions persist throughout, with dynamic Cu-O bond length variations. In-situ hard X-ray absorption and ex-situ soft X-ray absorption study, along with density function theory calculations, reveal two distinct charge compensation mechanisms at approximately 3.66 V and 3.99 V plateaus. Notably, we observe a Zhang-Rice-like singlet state during 3.99 V charging, offering an alternative charge compensation mechanism to stabilize the active oxygen redox reaction.

The development of next generation high energy density secondary batteries is critical to the realization of carbon-neutral goal by the mid-21st century. In the conventional intercalation type cathode (positive electrode) materials, which are widely used in the current Li-ion battery technology, the capacity is determined by the active redox of transition metal (TM) cations accompanying the extraction/insertion of $Li^+$ ions[1,2]. Researchers' attention was redirected to activating anion redox besides TM cation redox to harvest extra capacity[3–9]. From the first discovery of $Li_2MnO_3$ to the more recent development of cathode materials with the general formula of $Li[Li_xNi_yCo_zMn_{1-x-y-z}]O_2$ (the cations within the parenthesis reside in the TM layer), the Li-rich cathode materials have attracted broad attentions because of the excess capacity beyond TM redox[6–8,10–12]. Extensive studies have revealed that extra capacity is mainly from the activation of lattice oxygen redox reaction in these materials, which often show very high capacity with only small amounts of expensive elements such as Ni or Co[11,12]. They therefore make excellent candidates for next-generation cathodes. However, they suffer from large voltage hysteresis between charge/discharge and severe voltage decays during cycling, hindering their commercialization[13–15]. It is now generally agreed that the capacity at the charge plateau around 4.5 V (versus $Li^+$/Li) is mainly from the oxidation of lattice oxygen ions[16]. Yet, the exact nature of oxidized oxygen ions and the underlying reason for the subsequently large voltage hysteresis and voltage decay is still fiercely debated[16–18].

[1]Chemistry Division, Brookhaven National Laboratory, Upton, NY 11973, USA. [2]Institute of Physics, Chinese Academy of Sciences, 100190 Beijing, China. [3]X-ray Science Division, Advanced Photon Source, Argonne National Laboratory, Argonne, IL 60439, USA. [4]Department of Materials Science and Chemical Engineering, Stony Brook University, Stony Brook, NY 11794, USA. [5]Chemical Sciences Division, Oak Ridge National Laboratory, Oak Ridge, TN 37831, USA. [6]Neutron Scattering Division, Oak Ridge National Laboratory, Oak Ridge, TN 37922, USA. [7]Electrification and Energy Infrastructure Division, Oak Ridge National Laboratory, Oak Ridge, TN 37922, USA. [8]Energy Sciences Institute, Yale University, 810 West Campus Drive, West Haven, CT 06516, USA. ✉e-mail: enhu@bnl.gov; liuj1@ornl.gov

More recently, it is found that lattice oxygen redox reaction is also the dominant source of charge compensation at high degrees of charge in the conventional $LiTMO_2$ cathodes, such as those in $LiCoO_2$ and various Ni-rich cathodes[19,20]. Moreover, there are also evidences showing that the evolution of oxidized oxygen ions play an important role in the capacity degradation during long cycling[21,22]. It has been previously hypothesized that the oxidized oxygen ions can be partially stabilized through the p-d hybridization with neighboring TM cations[18,23]. However, this stabilization is found to be relatively weak, and oxidized oxygen ions tend to be released as oxygen gases and leads to the surface/subsurface reconstruction, a mechanism that has been broadly cited in various high energy density layered oxide cathodes (positive electrode materials)[15,21,22,24–26]. This motivates us to search for alternative mechanisms to stabilize oxidized lattice oxygen ions in order to achieve long-term reversible lattice oxygen redox reaction.

Anionic redox with small voltage hysteresis has recently been achieved in a handful layered sodium ion cathodes, e.g., $Na_2Mn_3O_7$ and P2/P3-type $Na_{0.6}Li_{0.2}Mn_{0.8}O_2$[27–34]. In these materials, the oxygen redox is activated by the A-O-A configuration where A represents a vacancy, an alkali cation, or a cation with strong ionic features such as $Mg^{2+}$ and $Zn^{2+}$. Presented with the A-O-A configuration, one of the oxygen's 2p orbitals is weakly hybridized with the neighboring TM cation which creates a unique narrow electronic state (band) with primary oxygen 2p features around the Fermi level, often referred as the "non-hybridized" or "orphaned" oxygen 2p states[9,32,33,35–37]. When the electrons are taken from these systems during charging, holes are created on these unique oxygen 2p dominated states instead of the conventional TM 3d-oxygen 2p hybridized states, and thus predominant lattice oxygen redox can be activated. Although small voltage hysteresis is observed in some of these materials, the retention of non-hysteresis capacity in long cycling is far from satisfactory[38–40]. The very small voltage hysteresis only pertains to the initial several cycles and it quickly enlarges during subsequent cycles. It was recently revealed that the O 2p holes created in these materials are stabilized via the localized resonant π bonding within the vacancy-$Mn_6/Ir_6$ rings, which effectively blocks both intra-layer (from original TM site to vacant site) and interlayer TM cation migration, leading to the observed non-hysteresis capacity[36,37]. It could be speculated that the rather weak structural stabilization (e.g., resistance to intra-layer TM cation migration) provided by the resonant p-d π-bonding is responsible for the quick fading of small voltage hysteresis. Exploring stronger stabilization mechanism for oxygen 2p electron holes may lead to strategies for achieving stable lattice oxygen redox in a more long-lasting way.

In this work, the recently discovered P3-type $Na_{2/3}Cu_{1/3}Mn_{2/3}O_2$ is used as a model material to demonstrate an alternative mechanism for non-hysteresis lattice oxygen redox reaction to be realized during long cycling, different from the more widely known p-d resonant π-bonding in $Na_2Mn_3O_7$ and P2/P3-type $Na_{0.6}Li_{0.2}Mn_{0.8}O_2$. The idea was raised based on that the p-d σ overlap is much stronger than the p-d π overlap (typically 1.7 – 2 times)[41]. The P-type Na-Cu-Mn-O system for sodium-ion cathode material was first introduced by Xu et al. in P2-type $Na_{0.68}Cu_{0.34}Mn_{0.66}O_2$ establishing the "$Cu^{2+}/Cu^{3+}$" redox utilization[42]. In the current P3-type $Na_{2/3}Cu_{1/3}Mn_{2/3}O_2$ material, through formation of σ overlap between O 2p holes and Cu 3d states, half of the $Na^+$ (the nominal oxidation of $Cu^{2+}$ to "$Cu^{3+}$") can be reversibly extracted/inserted at two different plateaus ( ~ 3.66 V and 3.99 V vs. Na/$Na^+$, voltages are all expressed vs. Na/$Na^+$ unless otherwise specified) with a small and stable voltage hysteresis during prolonged cycling. The formed hybridized states are in high resemblance to the "Zhang-Rice" singlet state in the hole-doped cuprates for high-temperature superconductors[43,44]. Detailed investigation on atomic structure and electronic structure revealed that at the 3.99 V charge plateaus, holes are predominantly created on a state with primary oxygen 2p parentage and it was stabilized by forming localized bound state with singly occupied Cu $dx^2-y^2$. The reversible capacity with near-zero voltage hysteresis in $Na_{2/3}Cu_{1/3}Mn_{2/3}O_2$ enlightens a route to stabilize long-term reversible oxygen redox reactions.

## Results and discussion

### Crystal structure of P3-type $Na_{2/3}Cu_{1/3}Mn_{2/3}O_2$ and Jahn-Teller distortion of $CuO_6$ octahedron

The as-prepared sample is fine black powder. High resolution synchrotron XRD (λ = 0.41285 Å) was collected (Fig. 1a). The diffraction pattern cannot be fully indexed using the conventional structure model of the P3-type cathodes (S.G. R3m, Supplementary Fig. 1a)[45]. Introducing the in-plane Cu/Mn honeycomb ordering and potential monoclinic distortion (S.G. Cm, as discussed previously for P3-type $Na_{2/3}Mg_{1/3}Mn_{2/3}O_2$) improves the fit substantially (Supplementary

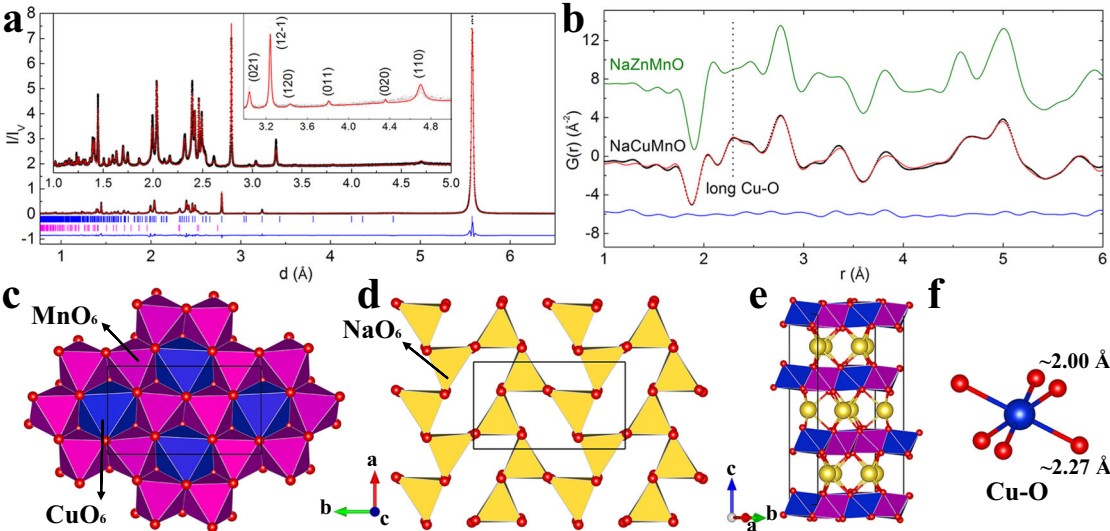

**Fig. 1 | Structural characterization of P3-type $Na_{2/3}Cu_{1/3}Mn_{2/3}O_2$ and the illustration of its structure.** Structure refinement of P3-type $Na_{2/3}Cu_{1/3}Mn_{2/3}O_2$ using **a** XRD (λ = 0.41285 Å) and **b** neutron PDF data. The experimental data are shown in black dots, calculated data in red curve and differences are shown in blue curve. The experimental PDF of P3-type $Na_{2/3}Zn_{1/3}Mn_{2/3}O_2$ (**b**) is shown in olive curve for comparison. Refined structures (from neutron diffraction data) of the pristine P3-type $Na_{2/3}Cu_{1/3}Mn_{2/3}O_2$ in top view of **c** $TMO_6$ layer and **d** $NaO_6$ layer, and **e** cell in side-view. **f** Illustrations of long and short Cu-O bonds in JT distorted $CuO_6$ octahedron.

Fig. 1b)[46]. However, a series of extra reflections still cannot be fully indexed using the latter structure and peak positions are also shifted away from the calculated model (Supplementary Fig. 1b). These discrepancies are likely routed in two aspects: first, strong local structural distortion induced by the Jahn-Teller (JT) effect of $Cu^{2+}$ ($d^9$); Second, in-plane Na-vacancy ordering, which has been previously observed in the P2-type $Na_{2/3}Ni_{1/3}Mn_{2/3}O_2$[45,47]. To fully resolve the structure, a full pattern indexing was carried out followed by the *ab* initio structure determination using the charge flipping method (details in Supplementary Fig. 2)[48,49]. Major reflections can be indexed using S.G. $P2_1/c$ with lattice parameters refined to be a = 6.6027(2) Å, b = 8.7137(2) Å, c = 5.0085(1) Å and β = 122.360(1)° (the few unindexed reflections are from CuO, Fig. 1a). $Na^+$ are found to reside in the interlayer prism sites and follow a herringbone ordering pattern, as can be seen in Fig. 1d. This specific arrangement maximizes the neighboring Na-Na distances by avoiding occupation of face-sharing prism sites. It also offers the lowest electrostatic repulsion between $Na^+$ and surrounding honeycomb-ordered TM cations (Fig. 1d).

This structure model was further refined using neutron diffraction data (Supplementary Fig. 3 and 4), which provides better sensitivity to oxygen ions and can also effectively distinguish Mn from Cu because of their very different coherent nuclear scattering lengths (−3.73 fm and 7.72 fm)[50]. The strong JT distortion of $Cu^{2+}$ ($d^9$) can be clearly seen from the refined average structure, with the two long Cu-O (×2) bonds refined to 2.271(4) Å while the two short pairs of Cu-O (×2) bonds refined to 1.999(2) and 2.009(5) Å (Supplementary Data 2). Superlattice reflections associated with the in-plane Cu-Mn honeycomb ordering cannot be fully modeled using the long-range ordered structure (S.G. $P2_1/c$). For instance, the observed intensity is much stronger than the calculated intensity for the 020 reflection (highlighted in Supplementary Fig. 3 and 4). These discrepancies are caused by the stacking disorder of Cu-Mn honeycomb-ordered layers, similar to those observed in $Li_2MnO_3$ or $NaNi_{2/3}Bi_{1/3}O_3$[51,52]. To confirm this, a numerical stacking disorder model was tested, details can be found in Supplementary Fig. 5. The best fit can be achieved by incorporating ~15% stacking disorder between successive honeycomb-ordered layers, a value close to that obtained from the phenomenological Cu/Mn anti-site disorder model (~15% anti-site defects) in the average structure (from POWGEN data, Supplementary Table 2). The splitting of two sets of Cu-O bonds and the in-plane Cu-Mn honeycomb ordering is further confirmed by the local structure refinement using short-range neutron pair distribution function (PDF) data (Fig. 1b, Supplementary Fig. 3 and Supplementary Table 2). The longer pairs of Cu-O bonds can be clearly seen, which are absent in the similar compound without JT cation (P3-type $Na_{2/3}Zn_{1/3}Mn_{2/3}O_2$, Fig. 1b). It is also worth noting that the longer Cu-O bonds (doubly occupied Cu $dz^2$) point toward the opposite direction for neighboring $CuO_6$ octahedra (along *b* axis, highlighted in Fig. 1c).

## Electrochemical performance of P3-type $Na_{2/3}Cu_{1/3}Mn_{2/3}O_2$ in half cell (versus Na/Na$^+$)

The theoretical full charging capacity of $Na_{2/3}Cu_{1/3}Mn_{2/3}O_2$ is 170 mAh/g if all 0.67 $Na^+$ ions/formula are extracted. However, complete extraction of $Na^+$ and oxidation of "$Cu^{2+}$" to "$Cu^{4+}$" are beyond the reach of the regular electrochemical test. Aside from this, oxygen redox may participate in the high voltage range and Mn redox may be active if excessive $Na^+$ is inserted at low voltage during discharge. To elucidate the full electrochemical characteristic of P3-type $Na_{2/3}Cu_{1/3}Mn_{2/3}O_2$, three different voltage windows were used for the electrochemical test, namely 4.5 V-2.0 V (Supplementary Fig. 6), 4.1 V-1.5 V (Supplementary Fig. 7), and 4.1 V-2.5 V (Fig. 2) (versus Na/Na$^+$, same for all the following voltages).

With 4.5 V and 2.0 V voltage cutoff, the charge and discharge capacity of the 1st cycle is 110 mAh/g and 85 mAh/g respectively (Supplementary Fig. 6a). During the initial charging, three major plateaus are presented at 3.66 V, 3.99 V and 4.40 V (Supplementary

Fig. 6b). The first charge plateau at 3.66 V has a capacity of 38 mAh/g corresponding to the removal of ~ 0.14 $Na^+$/formula, followed by the 3.99 V plateau at the end of which 80 mAh/g capacity is released and a total of 0.31 $Na^+$/formula is extracted (about half of all $Na^+$ ions). As for the 4.40 V plateau, despite a capacity of ~25 mAh/g during charge, no corresponding discharge capacity can be measured (Supplementary Fig. 6b), leading to a total discharge capacity of only 85 mAh/g contributed mainly from the 3.99 V and 3.66 V plateau (about 0.33 $Na^+$/formula insertion and a Coulombic Efficiency of mere 77%). Though the 3.66 V and 3.99 V plateaus are reversible, the reversibility quickly diminishes in the following cycles between 4.5 V and 2.0 V, as discharge capacity of these two plateaus become lower than 20 mAh/g after 30 cycles and new voltage peaks begin to emerge under 3.0 V (Supplementary Fig. 6b, c). The poor cycling performance indicates material degradation when cycled in the high voltage range, which could be induced by the irreversible reaction process at 4.40 V such as surface reconstruction or electrolyte decomposition.

In the wide voltage range between 4.1 V and 1.5 V, the charge/discharge voltage profile is much more reversible than that of 4.5 V-2.0 V (Supplementary Fig. 7). Beside 3.66 V and 3.99 V, a new reversible plateau appears just below 2.0 V which can be attributed to the $Mn^{3+}/Mn^{4+}$ redox. It is worth noting that the voltage of Mn redox here is much lower than that in other P-type sodium ion cathodes (e.g. Na-Fe-Mn-O) where Mn redox often occurs between 3.0 V and 2.0 V[53,54]. DFT calculation predicts a low potential ( ~ 1.5 V versus Na$^+$/Na) for Mn redox in the current Na-Cu-Mn-O system since the excessively sodiated state $Na_{5/6}Cu_{1/3}Mn_{2/3}O_2$ is high in total energy caused by inevitable face sharing between many $NaO_6$ prisms and $TMO_6$ octahedron (Supplementary Fig. 8). This difference of Mn redox occurring at lower voltage leads to a slightly lower capacity in the voltage range 4.1 V-2.0 V compared to the other reported state-of-art P3 type materials. For example, the initial discharge capacity of the current P3-$Na_{2/3}Cu_{1/3}Mn_{2/3}O_2$ material (between 4.1 V and 2 V) is only about half the capacity of that of P3-$Na_{0.9}Fe_{0.5}Mn_{0.5}O_2$ between 4.4 V and 1.5 V (79.8 mAh/g vs. 155 mAh/g)[53]. However, the initial discharge energy density of the P3-$Na_{2/3}Cu_{1/3}Mn_{2/3}O_2$ reaches 67% of that in the P3-$Na_{0.9}Fe_{0.5}Mn_{0.5}O_2$ (288.28 Wh/kg vs. 430 Wh/kg), due to the larger capacity and higher voltage associated with the 3.66 V and 3.99 V plateaus. Consequently, the presence of these two high-voltage plateaus is advantageous for the current material in terms of increasing energy density. More importantly, the following sections will demonstrate the mechanism related to them, showcasing the reversible utilization of oxygen redox with minimal voltage hysteresis.

Long-term cycling performance was tested in the voltage window of 4.1 V-2.5 V, as shown in Fig. 2. For the sample synthesized at 600°C through solid state reaction, discharge capacity retention is 63.2% after 100 cycles under the specific current of 10 mA/g. The CE is nearly 85% in 1$^{st}$ cycle and stabilizes at ~96% in prolonged cycling (Fig. 2a). Though the cyclability is not optimized, the sample was tested without any surface treatment or modification. Moreover, it is worth noting the excellent voltage stability and the nearly negligible voltage hysteresis between charge/discharge ( ~ 0.05 V, Fig. 2b, d). A closer inspection of the capacity of two separate plateaus (Fig. 2c) revealed that the capacity fading is mostly related the low CE of the higher voltage plateau at 3.99 V. When combined with the fact that no obvious voltage fading was observed after cycling, it suggests that no significant degradation occurs in the bulk material despite potential surface reconstruction after long cycling. Indeed, we found that synthetic procedure optimization can substantially enhance the overall cycling performance of this material. The cycling performances of three samples synthesized at different conditions under 50 mA/g are shown in Supplementary Fig. 9. Increasing the synthetic temperature from 600°C to 675°C can increase the overall charge capacity by about 5 mAh/g throughout the cycling. At 675°C, introducing trace amounts of Al-Mg-Ti dopants can also systematically improve the

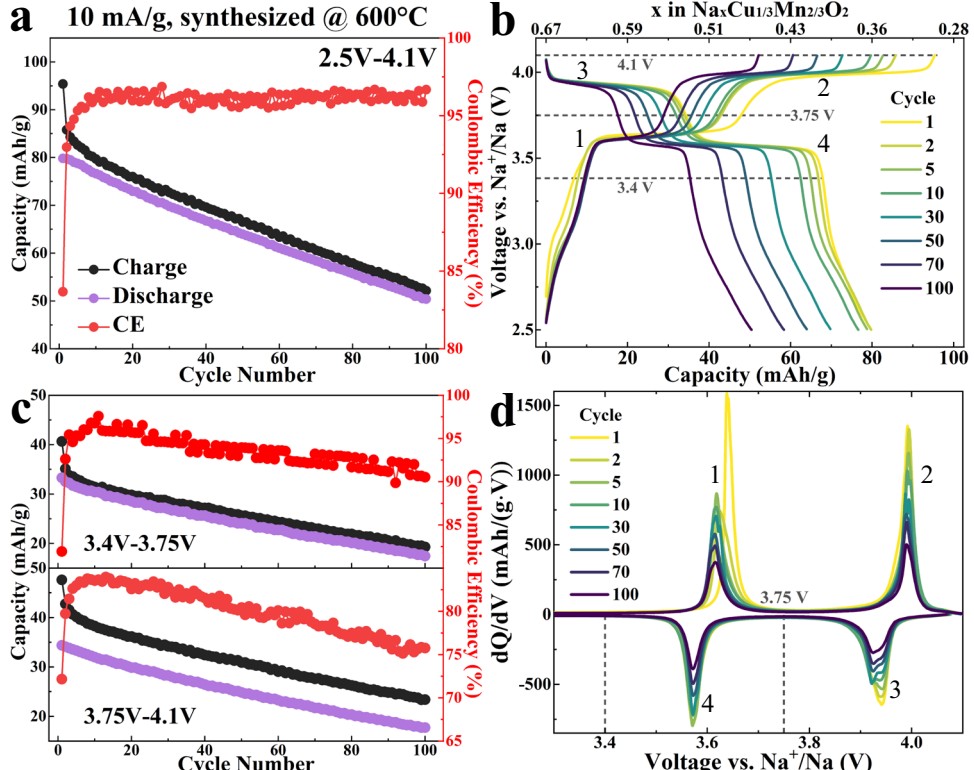

**Fig. 2 | Long cycling performance of 600°C-synthesized P3-type Na$_{2/3}$Cu$_{1/3}$Mn$_{2/3}$O$_2$ at 10 mA/g. a** Cycling performance of 600°C-synthesized P3-type Na$_{2/3}$Cu$_{1/3}$Mn$_{2/3}$O$_2$ sample with a specific current of 10 mA/g (between 2.5 V and 4.1 V for 100 cycles, versus Na$^+$/Na). **b** The typical charge and discharge profiles of P3-type Na$_{2/3}$Cu$_{1/3}$Mn$_{2/3}$O$_2$ at different cycles in the voltage window between 2.5 V and 4.1 V. **c** The charge, discharge capacity and the Coulombic efficiency (CE) change during 100 cycles in the voltage range around 1$^{st}$ 3.66 V plateau and 2$^{nd}$ 3.99 V plateau. **d** The differential voltage (dQ/dV) curves of the corresponding cycles in (**b**).

electrochemical performance. Compared to the 600°C sample, the overall charge/discharge capacity increases by about 10 mAh/g. The 1$^{st}$ cycle CE increases from 81.58% to 87.03%, and the discharge capacity retention increases from 70.82% to 73.45% after introducing trace doping. Further enhancement of cycling performance is expected if electrolyte optimization is adopted[55,56]. Regardless of the different synthetic optimizations, the two voltage plateaus at 3.66 V and 3.99 V are found to be stable and the voltage hysteresis between charge/discharge remains relatively small, all of which suggest that they are related to the intrinsic property of the P3-type Na$_{2/3}$Cu$_{1/3}$Mn$_{2/3}$O$_2$. In the following sections, the intrinsic stability is thoroughly investigated from the perspective of atomic structure and electronic structure.

## Structure evolution of P3-type Na$_{2/3}$Cu$_{1/3}$Mn$_{2/3}$O$_2$ during charge/discharge

The morphology and layer structure of the as-synthesized material were studied by high-resolution transmission electron microscopy (HR-TEM, Supplementary Fig. 10). The particle is about 1~2 μm in diameter and has perfect layer structure. Unfortunately, since the MnO$_6$ octahedra and CuO$_6$ octahedra stack together, the long and short Cu-O bond due to JT distortion cannot be distinguished by HR-TEM. To monitor the evolution of overall crystal structure and local atomic environment during electrochemical cycling, in-situ X-ray diffraction and absorption spectroscopy techniques were utilized.

High resolution in situ XRD data were collected for the initial charge and discharge. During the initial charging process, a two-phase reaction is identified for the 3.66 V plateau (Fig. 3a and Supplementary Fig. 11). The 100 and 200 reflections (in the monoclinic P2$_1$/c setting, corresponding to the 003 and 006 reflections in the conventional P3-type with S.G. R$\bar{3}$m) shift toward larger d-spacing, indicating the expansion of the Na$^+$ interlayer spacing. In contrast, the 10$\bar{2}$ and 20$\bar{2}$

reflections shift toward smaller d-spacing. These Bragg peaks reflect the dimension along the crystallographic c-axis. Interestingly, the 031 and 006 reflections, which are associated with the b-axis, remained almost unchanged. These observations suggest that lattice shrinks along the c-axis direction but remains nearly unchanged along the b-axis during the 3.66 V plateau. It is also worth pointing out that the superlattice reflection 12$\bar{1}$ disappeared while no new reflections emerged between (100)$_m$ and (200)$_m$ after this phase transition (Fig. 3a), implying that the decrease of Na$^+$ occupancy and the absence of new long-range Na-vacancy ordering. To fully resolve the structure of this partially desodiated phase, Rietveld refinement was carried out using both ex-situ high resolution XRD and TOF neutron diffraction data (Supplementary Fig. 12). Lattice parameters are refined to be a = 6.6255(6) Å, b = 8.7179(7) Å, c = 4.9708(4) Å and β = 122.028(4)°, which indeed agrees with the above qualitative description. Na$^+$ site occupancy (Wykoff 4e) is refined to be 0.72(6), indicating that around 0.48 Na$^+$/formula remained in the lattice after the initial phase transition. This is in reasonable agreement with the electrochemistry data where the capacity from the 3.66 V plateau corresponds to the removal of ~0.14 Na$^+$/formula.

During the 3.99 V plateau, major diffraction peaks remained unshifted despite the variation of corresponding intensities. A careful inspection reveals broad shoulders arise on the left side of the 100/200 reflection (highlighted in orange dotted line in Fig. 3a and Supplementary Fig. 11), suggesting there is a partial conversion of the P3-type phase to another phase with larger c lattice parameter. Due to the severe peak overlapping, it is difficult to quantitatively determine the structure in the data sets at the upcut-off voltage of 4.1 V. To determine the structure of this deep desodiated phase, in situ diffraction data were also collected in the voltage range of 4.5 V-2.0 V (Supplementary Fig. 11). Peak indexing of this new phase and structure determination (Supplementary Fig. 13a) suggests that the lattice oxygen ions in this

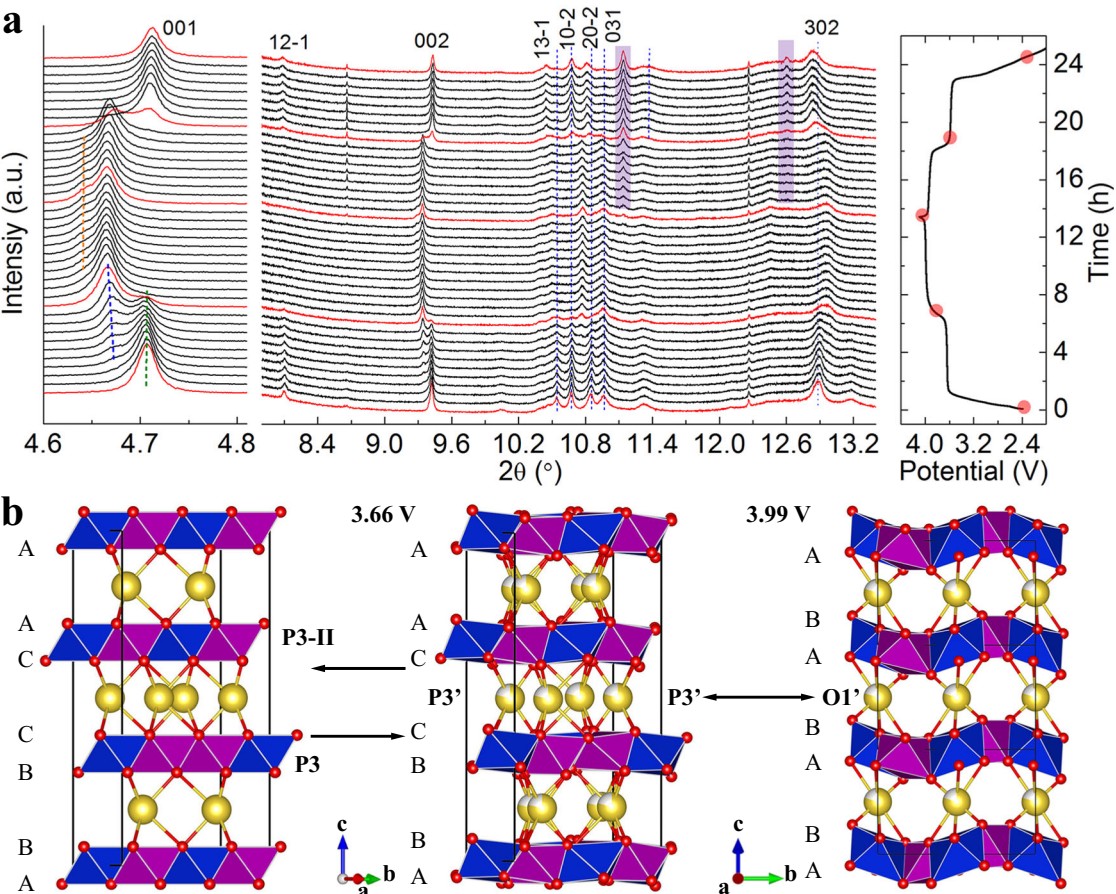

**Fig. 3 | Overall crystal structural evolution of P3-type Na$_{2/3}$Cu$_{1/3}$Mn$_{2/3}$O$_2$ monitored by in-situ XRD. a** In-situ XRD (λ = 0.457908 Å) of P3-type Na$_{2/3}$Cu$_{1/3}$Mn$_{2/3}$O$_2$ with up cut-off voltage of 4.1 V, the corresponding galvanostatic charge and discharge curve is shown on the right. **b** Illustration of the structure transition of P3-type Na$_{2/3}$Cu$_{1/3}$Mn$_{2/3}$O$_2$ during the initial charge/discharge.

desodiated structure follow the O1-type stacking (ABAB…, Fig. 3b). Na$^+$ are found to reside in the distorted NaO$_6$ octahedra and the amount of residual Na$^+$ is refined to be around 0.26 Na$^+$/formula in this phase. The charge capacity from the 4.4 V plateau is found to be not reversible, consistent with the irreversible structure transition when charged to 4.5 V (Supplementary Fig. 6). Therefore, the 4.1 V is the preferred cut-off voltage to efficiently utilize the 3.99 V plateau.

The structure transition is reversible during the 3.99 V discharge plateau (with 4.1 V cut-off), the O1' phase converted back to the P3' phase (Fig. 3b). However, the discharged P3' phase does not transform fully back to the original P3 phase during discharge past the 3.66 V plateau. Instead, it converts to a different P3-type phase (denoted as P3-II here) with a triclinic structure (S.G. *P*–1, a = 4.2443(5) Å, b = 5.2172 (3) Å, c = 11.6135 (4) Å, α = 103.775(3)°, β = 93.547(8)°, γ = 109.936(6)°, Supplementary Fig. 14). More interestingly, the P3' to P3-II phase transition occurs at a much fast rate ( ~ 30 min, or a single in-situ XRD scan, Fig. 3a) compared to the original P3 to P3' phase transition, where the two-phase co-exist region expands over six scans ( ~ 3 h). In the following cycles between 2.8 V and 3.1 V, the overall crystalline structure of the sample is shifting between this newly formed P3-II phase and O1' phase. Through prolonged cycling, the layer structured feature of the sample was well preserved with minimal transition metal migration into Na layer which agrees well with the non-voltage fading in the following cycles.

## Local Cu-O bond length dynamic change monitored by in-situ EXAFS (Extended X-ray Absorption Fine Structure)

Though the overall layer structure framework of P3-type Na$_{2/3}$Cu$_{1/3}$Mn$_{2/3}$O$_2$ material is preserved during the electrochemical cycle, the local structure around Cu atoms changes dramatically. As shown in the ex-situ nPDF profile of pristine material (Fig. 1b, f), the distorted CuO$_6$ octahedron feature is presented due to JT effects (Fig. 1f) caused by imbalanced electron occupation of two e$_g$ states for Cu$^{2+}$ (dz$^2$:2, dx$^2$-y$^2$: 1) in a six-fold O coordination (O$_h$ symmetry). To monitor the evolution of CuO$_6$ octahedron, time-resolved in-situ EXAFS characterization at Cu K-edge was performed during the 1$^{st}$ electrochemical cycle (Fig. 4). During the first voltage plateau of 3.66 V, the long Cu-O bond length barely changes with the variation being less than 0.01 Å. Meanwhile, the short Cu-O bond length experiences a much larger change: shrinking from ~1.98 Å to ~1.93 Å. Therefore, the initial desodiation process leads to a length decrease of the short Cu-O bond associated with the Cu dx$^2$-y$^2$ orbital while the long Cu-O bond associated with Cu dz$^2$ orbital remains almost intact. As a result, the JT distortion of CuO$_6$ octahedron becomes more severe. Right after the 3.66 V plateau, when the global structure has completed the transformation to P3' phase, the short Cu-O bond shrinks to about 1.92 Å and the long Cu-O bond slightly expands to over 2.31 Å. During the further desodiation at the second voltage plateau of 3.99 V, the evolution behaviors of both short and long Cu-O bonds follow the opposite trend to the situation at the 3.66 V plateau. The lengths of the short Cu-O bonds barely change while the long Cu-O bonds quickly increase to over 2.35 Å. The long Cu-O bonds associated with the Cu dz$^2$ orbital is much more affected at the 3.99 V plateau than the short Cu-O bond associated with Cu dx$^2$-y$^2$ orbital. This is in clear contrast to the initial Na$^+$ removal. In the subsequent discharge process, both long and short Cu-O bond lengths behave in the exact reverse manner as opposed to the

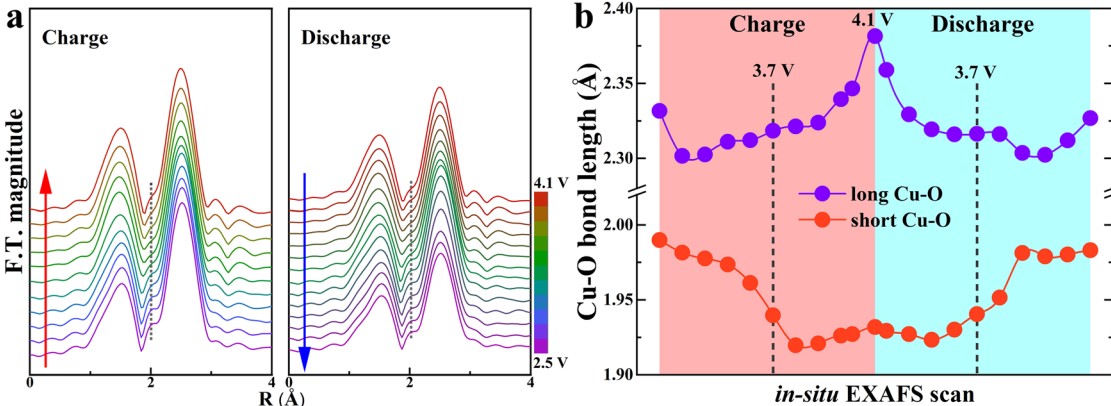

**Fig. 4 | Local Cu-O coordination change in the P3-type Na$_{2/3}$Cu$_{1/3}$Mn$_{2/3}$O$_2$ monitored by in-situ EXAFS. a** Fourier transformed spectra in real space of in-situ EXAFS at Cu K-edge between 2.5 V and 4.1 V with peaks indicating the Cu-O distance distribution in the 1$^{st}$ nearest shell. The red and blue arrow tell the order of acquisition for EXAFS data during the charging and discharging process. The dotted grey line marks the position of side-peak accompanying the Cu-O peak which indicates the preservation of short Cu-O bond and JT distortion. **b** Summarized bond length evolution of long and short Cu-O bond in the JT distorted CuO$_6$ octahedron during 1$^{st}$ cycle.

charging process. It is nearly fully recovered to the initial values of 2.33 Å and 1.98 Å respectively at the end of the initial discharge, demonstrating the highly reversible change of the local structure. It is consistent with the small voltage hysteresis and good cycling stability of this material. Taken together, these lead to two major observations: i) the JT distortion of CuO$_6$ octahedron is preserved throughout the charging/discharging process, and the length ratio between long/short Cu-O bond increases with Na$^+$ removal (as opposed to the conventional wisdom of forming Cu$^{3+}$); ii) detailed local structure evolution differs significantly at the 3.66 V and 3.99 V plateau, suggesting different charge compensation mechanisms since the JT distortion is dictated by the electron redistribution.

## Electronic structure evolution of P3-type Na$_{2/3}$Cu$_{1/3}$Mn$_{2/3}$O$_2$ characterized by in-situ hard XAS and ex-situ soft XAS

Besides the dynamic local structure evolution during electrochemical cycling, the chemical states evolution of transition metal and oxygen was monitored with synchrotron X-ray Absorption Near Edge Spectroscopy (XANES). Figure 5 shows the in situ XANES data at Cu K-edge during 1$^{st}$ cycle, ex-situ XANES data at Cu L-edge and O K-edge at different charging states. The Cu spectra show a typical rigid edge shift behavior during de-/sodiation cycle[57] and the shift is well reversible between charging and discharging which implies good electrochemical reversibility of Cu-related redox. During charging, the K-edge position of Cu continues shifting from 8987.55 eV at pristine state to 8988.29 eV at 4.1 V cut-off (Fig. 5a), which corresponds to the oxidation of Cu$^{2+}$ to Cu$^{3+}$ as reported before[57]. A closer inspection shows that about 0.44 eV edge shift happens during the 3.66 V plateau while only about 0.30 eV edge shift happens during the 3.99 V plateau. Such difference suggests different charge compensation mechanisms between the two plateaus and the contribution of Cu$^{2+}$/Cu$^{3+}$ redox to the second charging plateau of 3.99 V is likely very limited. Due to the covalent nature of Cu-O bond, the Cu and O electronic states are well hybridized. The more hybridization-sensitive Cu L-edge and O K-edge XANES data reveals delicate differences between electronic structure at the end of 3.66 V and 3.99 V plateaus.

At pristine state (Fig. 5c), the L$_3$-edge of Cu shows a nearly standard Cu$^{2+}$ feature with only one sharp white line peak around 930 eV (vertical dotted line in Fig. 5c) which corresponds to the transition from d$^9$ ground electronic configuration to the $\underline{c}$d$^{10}$ ($\underline{c}$ denotes a core hole). At the end of 3.66 V charging plateau, the white line peak at ~930 eV shifts towards higher energy, indicating the oxidation of Cu$^{2+}$ cation. Meanwhile, a new shoulder peak emerges just above 932 eV

(upward arrows in Fig. 5c) which is allegedly related to hybridization with ligand oxygen and oxygen hole states formation[43,58]. This interpretation is also supported by the O K-edge XANES data measured at the same time (Fig. 5d), where a shoulder peak (~526.5 eV, upward arrows in Fig. 5d) clearly emerges at the end of 3.66 V plateau before the main pre-edge peak (< ~528 eV, vertical dotted line in Fig. 5d) which has nearly zero shift from pristine state[59]. The changes of Cu L$_3$- and O K-edge XANES together suggest that, during 3.66 V plateau, Cu$^{2+}$ is oxidized and Cu-O hybridization becomes stronger. In the second charging plateau of 3.99 V (Fig. 5c), the white line peak around 930 eV of Cu L$_3$-edge XANES surprisingly moves back to the original position of the pristine state. Though it seems that Cu gets back to +2 valence state, the shoulder peak around 932 eV preserves and becomes even stronger with only a slight shift towards lower energy relative to that of the end of 3.66 V plateau, implying a different electronic state from that of standard Cu$^{2+}$ at pristine state. This new feature is closely related to ligand oxygen anions. As seen in Fig. 5d, both the main pre-edge peak ~528 eV and the shoulder peak ~526.5 eV slightly shift towards the higher energy end and the shoulder peak becomes more prevalent compared to that at the end of 3.66 V plateau, which suggests more participation of O in the active redox during the 3.99 V plateau. In fact, the electronic features observed here at end of 3.99 V charging plateau fit the profile of Zhang-Rice (ZR) singlet state that was first reported for high Tc superconducting cuprate materials[44,58,60-62]. The ZR state refers to a special correlated electronic state formed between a center Cu$^{2+}$ with one hole in its e$_g$ orbital and a ligand O also with one hole in its p orbital (Cu$^{2+}$:3d$^9$-O:2p$^5$ or simplified as d$^9\underline{L}$ where $\underline{L}$ denotes a ligand hole)[60-62]. The 530 eV white line peak and a shoulder peak around 531.5 eV-532 eV of Cu L$_3$-edge XANES are a signature of ZR singlet state corresponding to the excitation from d$^9\underline{L}$ to $\underline{c}$d$^{10}\underline{L}$[43,63]. Therefore, it is very likely that lattice oxygen dominates the redox reaction during 3.99 V plateau with a new correlated electronic state formed between O hole and Cu$^{2+}$ hole.

## Charge compensation mechanism analysis through DFT calculation

To gain a fundamental understanding of the unusual electronic state features observed for Cu and O, first principles spin-polarized DFT calculations were carried out for P3-type Na$_{2/3}$Cu$_{1/3}$Mn$_{2/3}$O$_2$ at different desodiation states. Results are summarized in Fig. 6 with a density of states (DOS) plotted. As the JT distortion of CuO$_6$ octahedron is rooted in the imbalanced electron distribution among Cu d orbitals hybridized with O p orbitals in coordination, DOS are projected (pDOS)

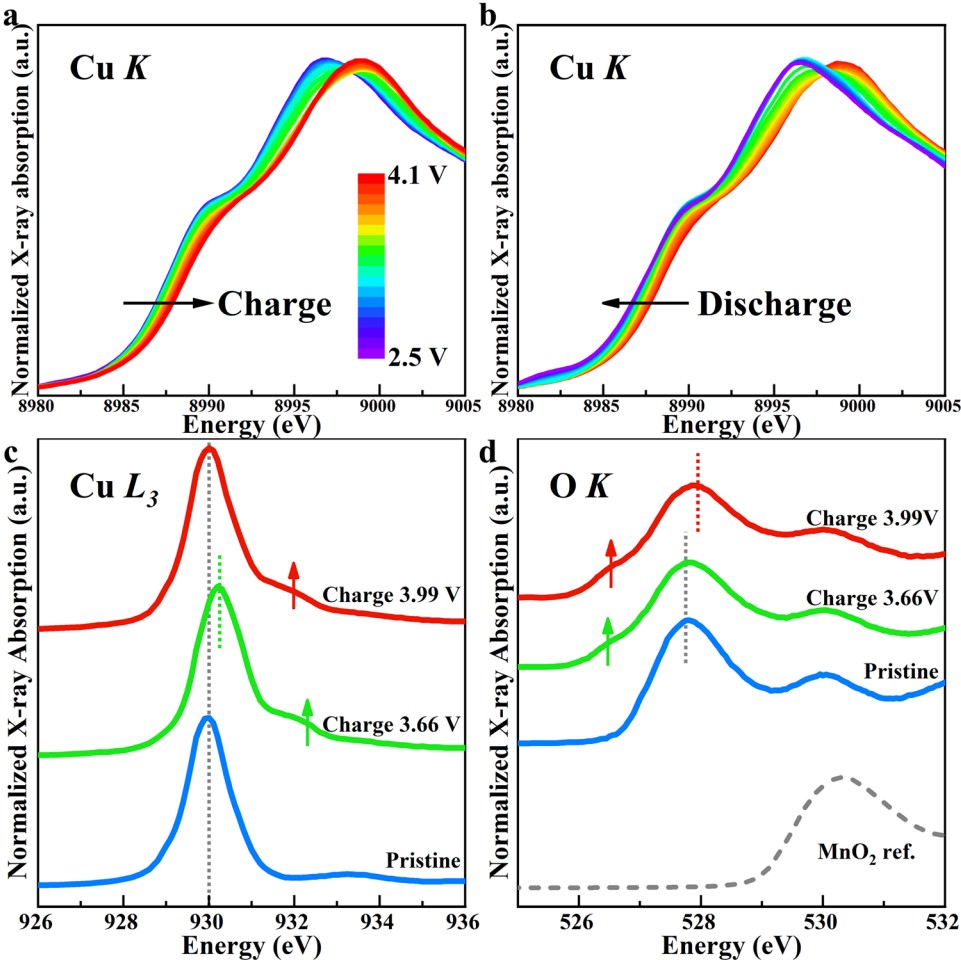

**Fig. 5 | Electronic structure change of P3-type Na$_{2/3}$Cu$_{1/3}$Mn$_{2/3}$O$_2$ and the ZR singlet-like features. a, b** In-situ XANES spectra at Cu K-edge during charging and discharging of 1$^{st}$ cycle. Ex situ XANES spectra at **c** Cu L$_3$ edge and **d** O K-edge for the pristine state, end of 3.66 V plateau, and end of 3.99 V plateau. The vertical dotted lines indicate the position of the main peak, and the upward arrows indicate the rising of shoulder peaks and their positions.

onto O-2p and Cu-3d states in Fig. 6. Furthermore, the DOS of models with and without JT distortion (Fig. 6a) are compared to associate the CuO$_6$ distortion with electronic features directly (details of model construction can be found in methods and SI).

First, for P3-type Na$_{2/3}$Cu$_{1/3}$Mn$_{2/3}$O$_2$ at all desodiation states, obvious band gaps exist, indicating electronic insulation in consistency with the localized electronic feature of Mn$^{4+}$-populated oxides. On the ends of the gap are the Fermi level (top of the valence band, ToV) and the bottom of conduction band (BoC), which dynamically change during desodiation and consist of states related to active redox[64]. In the pristine material, the typical electronic feature of Cu$^{2+}$ (d$^9$) cation in a six-fold oxygen coordination is presented (Fig. 6b). Dictated by the O$_h$ symmetry of O$_6$ octahedron, the originally degenerated five d orbitals split into t$_{2g}$ states (dxy, dyz, and dxz) and e$_g$ states (dx$^2$-y$^2$ and dz$^2$). As shown in Fig. 6b, the e$_g$ antibonding states take over the ToV. More specifically, the electronically filled e$_g$* states at ToV consist of dz$^2$ orbital in both spin-up and spin-down channels and dx$^2$-y$^2$ orbital in the only spin-up channel. Such imbalance causes larger repulsion along z-axis of CuO$_6$ octahedron resulting in an elongated Cu-O bond in z direction (Fig. 6a, also observed by nPDF and EXAFS). Furthermore, the pDOS of O 2p at ToV are clearly in accordance with the pDOS of Cu which suggests the hybridized nature of e$_g$* states at Fermi level. Meanwhile, almost no Mn-3d states are in the vicinity to ToV (Supplementary Fig. 15). These electronic structure features imply that, at the beginning of charge, Cu$^{2+}$ cation is electrochemically active and its hybridized e$_g$*

states with oxygen are solely responsible for the observed charge compensation.

At the end of 3.66 V charging plateau (Na$_{1/2}$Cu$_{1/3}$Mn$_{2/3}$O$_2$), new empty bands emerge in the middle of the band gap (highlighted region in Fig. 6c, d). For both the situations with and without CuO$_6$ JT distortion, the newly emptied band is dominated by Cu e$_g$* states and shows a strongly hybridized feature since the O 2p states are in high accordance. However, a significant difference exists in the detailed composition of this empty band between the situations with and without CuO$_6$ JT distortion. When the octahedron distortion is released (Fig. 6c), the empty band consists of degenerated Cu dz$^2$ and dx$^2$-y$^2$ states in the spin-down channel. The empty dz$^2$ states originate from the electron depletion of the originally filled spin-down dz$^2$ orbital at ToV in the pristine material (Fig. 6b) while the empty dx$^2$-y$^2$ states are separated from the already emptied spin-down dx$^2$-y$^2$ orbital at the pristine state. Non-distort CuO$_6$ octahedron creates a symmetrical crystal field that equalizes energy levels of dz$^2$ and dx$^2$-y$^2$ making them degenerate. In contrast, when JT distortion is preserved in the model (Fig. 6d), the empty band is almost completely composed of Cu dx$^2$-y$^2$ states in the spin-up channel which corresponds to electron depletion of originally filled spin-up Cu dx$^2$-y$^2$ states in pristine material (Fig. 6b). Such depletion would only worsen the imbalance between z direction and xy direction causing more severe octahedron distortion, in consistency with the EXAFS result (Fig. 4). This also explains why the short Cu-O bond related to Cu dx$^2$-y$^2$ orbital is mainly affected by desodiation at the 3.66 V plateau. Due to the preserved asymmetrical

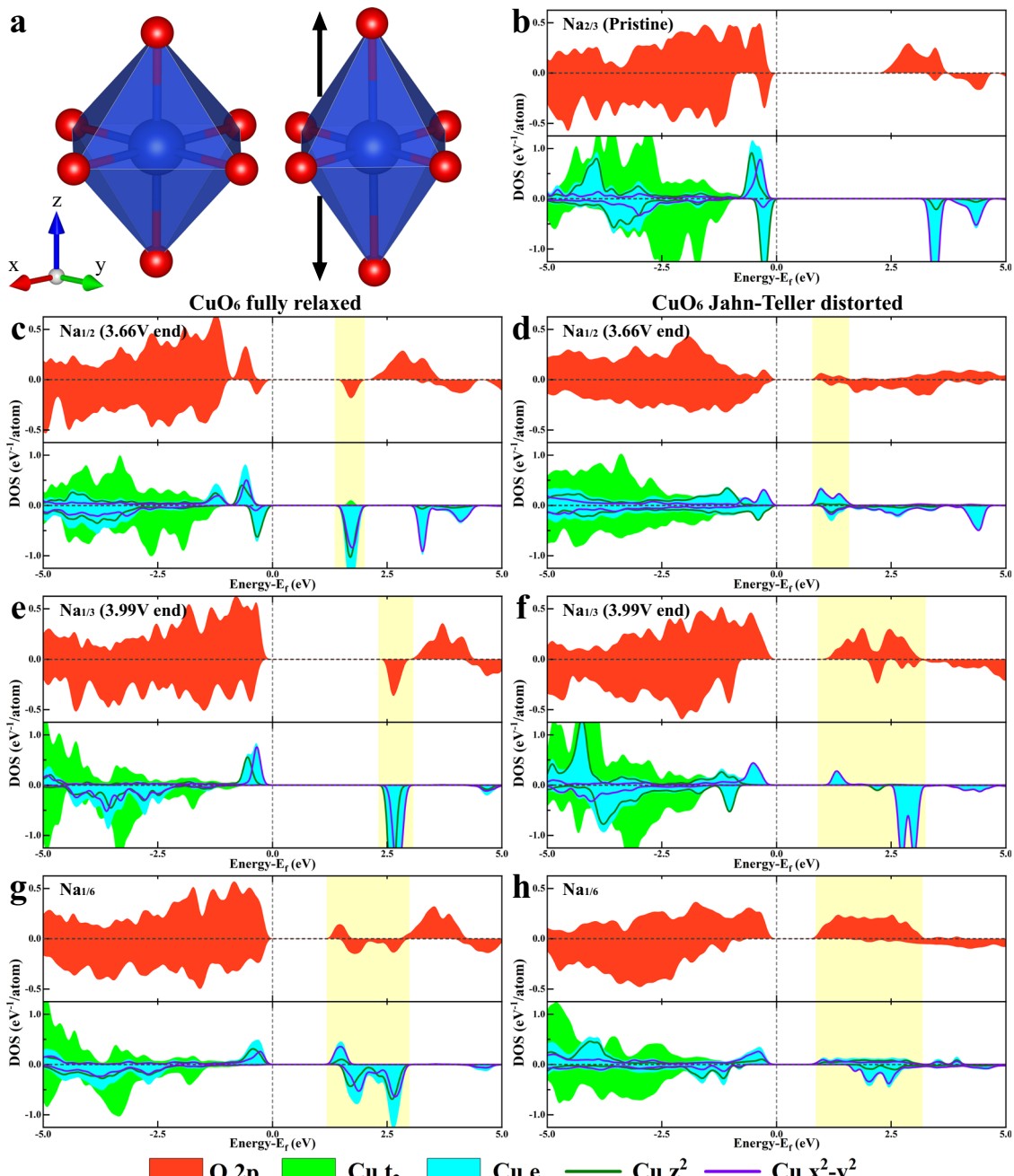

**Fig. 6 | DFT-calculated pDOS of P3-type Na$_{2/3}$Cu$_{1/3}$Mn$_{2/3}$O$_2$ at different desodiation state. a** Illustrations of fully-relaxed and JT-distorted CuO$_6$ octahedron. **b–h** DFT calculated pDOS of Na$_{2/3-x}$Cu$_{1/3}$Mn$_{2/3}$O$_2$ model at: **b** x = 0 pristine state, **c, d** x = 1/6 end of 3.66 V plateau, **e, f** x = 1/3 end of 3.99 V plateau, and **g, h** x = 1/2 at higher voltage than 4.1 V. pDOS of model structures with fully relaxed and JT-distorted CuO$_6$ octahedron are compared in parallel.

crystal field, the dz$^2$ orbital and dx$^2$-y$^2$ orbital are still split in energy. Taking together, the Cu$^{2+}$ oxidation dominates the electron depletion behavior throughout the 3.66 V plateau while ligand O anion mainly serves as an electron donor spectator. Besides, since the JT distortion is observed to maintain, most of the electrons should be extracted from the Cu dx$^2$-y$^2$ orbital leaving behind hole states at Cu site in strong hybridization with O 2p states.

Moreover, after deep desodiation at 3.99 V plateau (Na$_{1/3}$Cu$_{1/3}$Mn$_{2/3}$O$_2$), the mid-gap empty band extends in energy and becomes more significant (highlighted region in Fig. 6e, f) than the first plateau of 3.66 V. More importantly, after 3.99 V plateau in the JT distortion-preserving case, the oxygen states make a dominant contribution to the mid-gap empty band different from the situation after

3.66 V plateau (Fig. 6f). The Cu spin-up dx$^2$-y$^2$ state only takes a small portion of the empty mid-gap band at BoC while the majority of it is still filled dominating the ToV. This feature implies Cu$^{2+}$ is only partially oxidized but not fully oxidized to the nominal valence state of "Cu$^{3+}$". The large amount of empty O states at BoC suggests that, with CuO$_6$ octahedron JT distortion preserved, at 3.99 V plateau most of the electron extraction happens to O orbitals, leaving behind oxygen holes states. However, such oxygen holes are not necessarily related to isolated O$^{2-}$ oxidation. As can be seen in Fig. 6f, at BoC the O 2p states are still in high accordance with Cu e$_g$* states implying strong Cu-O hybridization. Furthermore, the originally emptied spin-down dx$^2$-y$^2$ states of Cu$^{2+}$ cation shift down in energy becoming energetically overlapping with the newly formed oxygen holes states in spin-up

channel which indicates new correlation formed between them. Such an anti-ferromagnetically coupled ligand O hole and Cu ($d^9$) hole state is the typical ZR singlet electronic configuration which validates the previous assumption of ~932 eV shoulder peaks observed in Cu $L_3$-edge XANES (Fig. 5c)[43,44,65]. In high Tc superconducting cuprates, oxygen holes are intentionally introduced through aliovalent doping. In currently studied P3-type $Na_{2/3}Cu_{1/3}Mn_{2/3}O_2$, large amounts of $Na^+$ removal at high voltage may create similar doping conditions. The much-elongated Cu-O bonds along $z$ direction (see EXAFS results) creates a local $CuO_4$-like planar ligand environment similar to the situation in superconducting cuprates. This ZR-like state also helps explain the stabilized short Cu-O bond length observed in EXAFS results since the ZR singlet state is energetically stable due to strong electronic correlation. Meanwhile, as electron depletion proceeds during 3.99 V plateau, the imbalance between $z$ direction and $xy$ direction worsens which find outlet by further stretching the long Cu-O bond along $z$ direction. In contrast, Fig. 6e shows the electronic structure model without $CuO_6$ JT distortion after 3.99 V plateau. The charge compensation mechanism is identical to that of 3.66 V plateau which is the electron depletion from originally filled Cu $d_{z^2}$ orbitals and O serves as a ligand in hybridization. The newly formed empty mid-gap states consist of degenerated Cu $d_{z^2}$ and $d_{x^2-y^2}$ states. Finally, further desodiation to $Na_{1/6}Cu_{1/3}Mn_{2/3}O_2$ does not change the orbital composition of electronic states at ToV and BoC calculated in $Na_{1/3}Cu_{1/3}Mn_{2/3}O_2$ but only change the energetic distribution of each orbital (Fig. 6g, h). Therefore, further electron depletion after 3.99 V plateau follows the same mechanism as that during 3.99 V plateau. However, with the current experimental cell testing set up, the reversible electrochemical process above 4.1 V cannot be realized. Future works on cell optimization and electrolyte engineering may be needed to validate the prediction from DFT calculation.

In summary, the spin singlet state in the P3-type $Na_{2/3-x}Cu_{1/3}Mn_{2/3}O_2$ can be effectively used to achieve the non-hysteresis and long-term stable oxygen redox reaction. High resolution in situ XRD reveals highly reversible structure transitions during charge/discharge. Local $CuO_6$ JT distortion are preserved throughout electrochemical cycling with dynamic Cu-O bond length evolution. In situ hard X-ray absorption and ex situ soft X-ray absorption study, coupled with density function theory calculation reveals two different charge compensation mechanisms for ~3.6 V and ~4.0 V plateaus. Particularly, a Zhang-Rice-like singlet state is observed in this compound during charge, providing an alternative charge compensation mechanism with stable active oxygen hole participation. This finding opens a plausible route to explore high energy density cathodes with reversible lattice oxygen redox reaction.

## Methods

### Material synthesis
All materials were synthesized using solid-state method. In a typical reaction, a stoichiometric precursor of sodium nitrate ($NaNO_3$, 2.39 g), Manganese carbonate ($MnCO_3$, 3.22 g), and copper nitrate trihydrate ($Cu(NO_3)_2 \cdot 3H_2O$, 3.35 g) are grinded in a mortar and pestle for 5 minutes before preheating at 500 °C for 12hrs in a muffle furnace. The preheated products were grinded in mortar and pestle again before heat treatment at different temperatures. For P3-type $Na_{2/3}Cu_{1/3}Mn_{2/3}O_2$ material, temperatures of 600 °C and 675 °C were chosen to obtain samples with different crystallinity.

P3-type $Na_{2/3}Cu_{1/3}Mn_{2/3}O_2$ material was also surface coated with a composition of 1 wt% of Al, 1 wt% of Mg, and 1 wt% of Ti. For a batch synthesis, sodium nitrate ($NaNO_3$, 2.39 g), Manganese carbonate ($MnCO_3$, 3.22 g) and copper nitrate trihydrate ($Cu(NO_3)_2 \cdot 3H_2O$, 3.35 g) were weighted, which were the same with the non-coated material. The source of Al, Mg, and Ti are Aluminum hydroxide ($Al(OH)_3$, 0.0127 g), Magnesium nitrate hexahydrate ($Mg(NO_3)_2 \cdot 6H_2O$, 0.0464 g) and Titanium(IV) butoxide ($Ti(OC_4H_9)_4$, liquid form, 0.0446 g). Similarly, all the precursors were grinded in a mortar and pestle for 5 minutes

before the preheat treatment at 500 °C for 12hrs. The final P3-type AMT-coated $Na_{2/3}Cu_{1/3}Mn_{2/3}O_2$ material was synthesized via a second heat treatment in a muffle furnace at 675 °C.

### Electrochemistry and ex-situ sample preparation
The cathode (positive) slurry electrode was made from a uniform slurry of active materials, acetylene black, polyvinylidene fluoride (PVDF) in N-methyl-2-pyrrolidone (NMP), with a ratio of 8:1:1 respectively. The resulting slurry was casted on Al foil, which was dried at 120 °C under vacuum overnight. CR2032 coin cells were assembled against the Na metal in an Ar-filled glovebox to evaluate the electrochemical performances. 1.2 Molar $NaClO_4$ in Propylene carbonate (PC) with 2% Fluoroethylene carbonate (FEC) was used as an electrolyte. The half cells (with excess Na metal and electrolyte) were then cycled at various conditions according to the purpose of testing. When assembling half cells, sodium cubes (Sigma-Aldrich) were first washed by a dimethyl carbonate (DMC, Sigma-Aldrich) solution to remove the mineral oil on the surface, and then sandwiched in between two plastic sheets to be rolled into a foil with a thickness of around 0.8 mm. Then sodium disks with a diameter of 12 mm were punched out before being put into the coin cells. The electrolyte volume in each coin cell is around 0.09 mL to ensure good wettability towards the glass fiber separator. For the first 2 cycles between 4.1 V and 1.5 V, the half cell was tested at 10 mA/g with an active material mass loading of 5 mg/cm² in the electrode. For the cycling comparison between 4.5 V-2.0 V and 4.1 V-2.5 V, the half cells were tested at 10 mA/g with an active material mass loading of 3 mg/cm². For the long-term cycling performance comparison between materials synthesized with different optimization methods, half cells were tested at 50 mA/g with an active material mass loading of 3 mg/cm². All cell testing was done in an air-conditioned room with little temperature variation (25 °C ± 2 °C).

### In situ and ex situ synchrotron X-ray diffraction
Ex-situ high-resolution synchrotron X-ray powder diffraction data were collected at beamline 11-BM ($\lambda = 0.41285$ Å) at the Advanced Photon Source, Argonne National Laboratory. Samples were loaded in 0.8 mm Kapton capillaries (Cole-Parmer; 1/32 inch ID and 1/30 inch OD).

*Operando* X-ray diffraction studies were performed at beamline 11-BM ($\lambda = 0.457908$ Å) at the Advanced Photon Source, Argonne National Laboratory in AMPIX cells[66]. Electrodes were prepared and cells were assembled in an Ar-filled glovebox. Working electrodes were 10 mm pellets (around 23 mg in total mass) pressed under uniaxial pressure consisting of active material, graphite, carbon black (Vulcan XC 72 R), and PTFE (Sigma-Aldrich) in a ratio of 60:10:10:20 (wt %), respectively. Cells consisted of a Na metal foil as a combined counter and reference electrode, a glass fiber separator, and an electrolyte of 1 M $NaClO_4$ in propylene carbonate (PC) with 2 wt. % FEC as additive. Cells were placed in a motor-controlled sample holder to switch between cells during experimentation and were cycled galvanostatically in the beamline using a Maccor 4300 battery cycler between 2.0 V and 4.1 V or 4.5 V at C/15 rate (specific current of 5 mA/g).

### Ex situ neutron diffraction
The neutron scattering was measured at the NOMAD beamline at SNS, ORNL. 3 mm thin-walled quartz capillary is used as sample holder[67,68]. Four 24 min scans (2 C proton charge each) were collected and summed together to improve the statistics. To obtain ~150 mg material at different states of charge, both coin cells with pellet electrode and pouch cells were used. To avoid the neutron signal interference from H in PVDF, the Polytetrafluoroethylene (PTFE) is used as binder in the electrode. The composition of the pellet is active material: Super P carbon: PTFE binder = 90:5:5 in weight ratio. The pellet was pressed using hydraulic press at 3 MPa pressure and ending with half inch diameter and thickness of about 120 μm. Similarly, the slurry coating

for pouch cells is made with the same composition. The hard carbon slurry electrode with the composition of 85:15:5 (weight ratio of hard carbon, Super P and PVDF binder) is used as the anode in the pouch cell. 10 cm × 10 cm pouch cells were made in the dry room using hard carbon as the anode. Scaled reduced neutron pair distribution function $G(r)$ was obtained by Sine Fourier transform of the normalized scattering function according to the following formula with a $Q_{max}$ of 30 Å$^{-1}$:

$$G(r) = (1 - g(r))r = \frac{1}{2\pi^2 \rho_0} \int_{Q\min}^{Q\max} Q(S(Q) - 1)Sin(Qr)dQ \quad (1)$$

Pattern indexing, structure solution, and structure refinements were carried out in the TOPAS v6 software[69]. For structure refinements using synchrotron XRD data, a fundamental peak profile approach was used to describe the instrument peak profile of 11-BM. The sample broadening from size and microstrain was also included during the refinement. Because of the presence of stacking disorder, we found it necessary to use the phenomenological strain broadening model to model the anisotropic broadened peaks.

For structure refinements using neutron Bragg diffraction data, time-of-flight (TOF) data were converted to d-spacing data using the second-order polynomial TOF = ZERO + DIFC*d + DIFA*d$^2$, where ZERO is a constant, DIFC is the diffraction constant. During the structure refinement, ZERO and DIFC were determined from the refinement of a standard NIST Si-640e data set and held fixed, while DIFA was allowed to vary during refinements to account for sample displacement. For low-angle banks (bank2 and 3, with center $2\theta = 31°$ and $67°$), a back-to-back exponential function convoluted with a Pseudo-Voigt function was used to describe the peak profile. For high resolution back scattering banks (banks 4 and 5, with center $2\theta = 122°$ and $154°$), the moderator-induced line profile was modeled using a modified Ikeda-Carpenter-David function[70–72]. Lorenz factor is corrected by multiplying d$^4$.

For the small box structure refinement using neutron PDF data. A Lorentzian dampening function (Exp(- r*Q$_{damp}$/2)) and Pseudo-Voigt-like peak profile were used to model the reduced $G(r)$ data in TOPAS v6. An empirical (-δ/r$^2$) term was used to describe the sharpening of low-r peaks due to the correlated atomic motion.

## Ex situ and in situ hard X-ray absorption spectroscopy

Cu, Mn (Supplementary Fig. 16) in-situ K-edge XAS measurements were performed at QAS (7-BM) beamline of the National Synchrotron Light Source II (NSLS II) at Brookhaven National Laboratory (BNL) in the transmission mode. The XANES and extended X-ray absorption fine structure (EXAFS) spectra were processed using the Athena software package (Supplementary Figs. 17–19).

## Soft X-ray absorption spectroscopy

Samples for ex-situ soft X-ray absorption were made from electrodes in coin cells. Five cells were assembled with the same electrode and electrode and were charged to 40 mAh/g (about 3.8 V), 4.1 V, 4.65 V, discharged to 3.75 V, and discharged to 2.0 V respectively. The electrodes were cleaned thoroughly with dried DMC solvent in the glovebox before the measurement. O K-edge and Cu L-edge XAS were collected by using either total electron yield (TEY) or partial fluorescence yield (PFY) modes at the IOS (23-ID-2) beamline of NSLS II, BNL (Supplementary Fig. 20). PFY spectra were collected using a Vortex silicon drift detector. All of the XAS measurements were performed at room temperature in an ultrahigh-vacuum chamber (base pressure, ≈10$^{-9}$ Torr).

## Model structure preparation for theoretical study

To investigate the electronic structure change in P3-type Na$_{2/3}$Cu$_{1/3}$Mn$_{2/3}$O$_2$ that happens during two charging/discharging plateau, we

prepared four structural models corresponding to different desodiation states of Na$_{2/3-x}$Cu$_{1/3}$Mn$_{2/3}$O$_2$, namely x = 0, 1/6, 1/3, 1/2 (Supplementary Figs. 21–24). For the pristine model of x = 0, in-plane Cu-Mn and Na-Vacancy arrangements were determined from the results of NPD refinement as shown in Fig. 1. For models of x = 1/6, 1/3 and 1/2, the in-plane Cu-Mn configuration was kept the same as that in pristine state since no Cu-Mn disorder and TM migration was observed in the bulk structure as suggested by NPD and in-situ XRD refinement results. The in-plane Na-Vacancy arrangement needs to be determined. For x = 1/3, the refinement result of in-situ XRD data suggests an ordered Na-Vacancy configuration as shown in Supplementary Table 7, which was used directly in DFT calculation. Meanwhile for x = 1/6 and 1/2, no obvious in-plane ordering was implied by refinement results. Therefore, firstly we generated all the inequivalent Na-Vacancy arrangements in a 2×2×2 supercell containing 48 units formula of Na$_{2/3-x}$Cu$_{1/3}$Mn$_{2/3}$O$_2$ using the enumeration method developed by Hart et al. [73,74]. Then we approximated the stable ground-state structure by first ranking all arrangements by their Ewald electrostatic energies and took the first thirty lowest energy configurations for a second round of DFT-based structure optimization. The ones with the lowest DFT energy were adopted as the structural model for Na$_{2/3-x}$Cu$_{1/3}$Mn$_{2/3}$O$_2$ at x = 1/6 and 1/2. Besides desodiated states, the Na$_{5/6}$Cu$_{1/3}$Mn$_{2/3}$O$_2$ state was also modeled to estimate the average voltage of Mn-redox activation during excessive sodiation. The in-plane Na-Vacancy configuration was searched for in a cell of the same size as that for pristine state Na$_{2/3}$Cu$_{1/3}$Mn$_{2/3}$O$_2$ where all prismatic sites were considered including both edge sharing and face sharing ones and 56 different Na-Vacancy configurations were compared for the one with the lowest DFT energy to be chosen as approximation of ground state. The DFT energies used in all the above model screening steps were evaluated within GGA + U framework the details of which are described in the following section of the method.

## DFT calculation details

All DFT calculations were performed with the Vienna ab initio simulation package (VASP) with projector augmented wave (PAW) approach[75,76]. For the structural relaxation in the model structure construction process described above, out of the consideration of balancing accuracy and computational cost, the generalized gradient approximation (GGA) type exchange–correlation functional in the parameterization by Perdew, Burke, and Ernzerhof (PBE) was adopted[77] in a GGA + U framework. U value of 5.0 eV and 4.0 eV were used for Cu and Mn respectively[43,78,79]. For the calculation of electronic structure for the selected ground-state model structure, the Heyd–Scuseria–Ernzerhof (HSE) hybrid functional was adopted, which is reported to give more accurate results than the common DFT + U functional, especially in describing empty states above Fermi level[80,81]. For HSE type functional, the choice of mixing parameter is important since it directly determines how much electron exchange is considered in the calculation, and is closely related to the energy level of empty states. A rigorous approach to obtaining mixing parameters is nonempirically tuning the fraction of mixing as suggested by Kronik and co-workers[82,83]. In this work, as we focus on the relative change of the main electronic feature of Na$_{2/3}$Cu$_{1/3}$Mn$_{2/3}$O$_2$ during desodiation rather than the exact energy level of electronic states, the standard mixing parameter of 0.25 for HSE06 type functional was adopted. For all calculations, the cutoffs of the plane-wave function basis was 500 eV. A k-mesh with a density of one point per ≈0.03 Å$^{-3}$ was generated using the Monkhorst– Pack method to ensure the precision of the calculated total energy. For the relaxations of model structures, the forces felt by each of the atoms were well converged below 0.01 eV Å$^{-1}$.

To compare the electronic structures of the model with a fully relaxed CuO$_6$ octahedron and JT distorted CuO$_6$ octahedron, we adopted two different structural relaxation approaches. We found that, for any desodiated state, the DFT + U functional used cannot

properly described the interaction between oxidized $Cu^{2+}$ and $O^{2-}$ and would result in a fully relaxed $CuO_6$ octahedron with six equally long Cu-O bonds. The DFT + U relaxed structures were then used as the models with fully relaxed $CuO_6$ octahedron whose electronic structure was further evaluated with HSE functional. To mimic the $CuO_6$ JT distortion in DFT model structure as observed by PDF and EXAFS, we manually set the atomic coordination to the values obtained from the refinement of diffraction data and kept the cell shape and size same as that from DFT + U relaxation. Then in the built model structure, only Na atoms were allowed to relax while all TM and O atoms were kept at the coordination from refinement. Since the DFT + U relaxed cell size was quite close to the measured one, the $CuO_6$ JT distortion and local environment of Cu and O from measurement were essentially recreated in the DFT model (detailed codes/results in Supplementary Data 1).

## Reporting summary

Further information on research design is available in the Nature Portfolio Reporting Summary linked to this article.

## Data availability

All data supporting this study and its findings are available within the article and Supplementary Information. Additional supporting data of this study are available from the corresponding author on request.

## Code availability

Code for DFT calculation is available in the supporting information. Codes for all neutron/X-ray data reduction are open source and are available at corresponding instruments.

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

## Acknowledgements

The work done at Brookhaven National Laboratory is supported by the Assistant Secretary for Energy Efficiency and Renewable Energy, Vehicle Technology Office of the US Department of Energy through the Advanced Battery Materials Research (BMR) Program under contract no. DE-SC0012704. Neutron powder diffraction measurements used resources at the Spallation Neutron Source (POWGEN and NOMAD instruments), a DOE Office of Science User Facility operated by the Oak Ridge National Laboratory. This research used resources of beamlines 7-BM (QAS) and 23-ID-2 (IOS) of the National Synchrotron Light Source II (NSLS-II) at Brookhaven National Laboratory (Contract No. DE-SC0012704 and DE-SC0012653). We acknowledge the use of facilities within the Eyring Materials Center at Arizona State University supported in part by NNCI_ECCS-1542160. The DFT calculations were performed using computational resources at the Center for Functional Nanomaterials (CFN), which is a U.S. Department of Energy Office of Science User Facility, at Brookhaven National Laboratory. The authors also acknowledge the Beijing Super Cloud Center (BSCC) and the Beijing Beilong Super Cloud Computing Co. Ltd for providing HPC resources that have contributed to the DFT results reported within this paper (http://www.blsc.cn/) This research used 11-BM beamline and electro-chemistry laboratory resources of the Advanced Photon Source, a U.S. Department of Energy (DOE) Office of Science User Facility operated for the DOE Office of Science by Argonne National Laboratory under Contract No. DE-AC02-06CH11357. J.L. would like to thank partial finacial support from ORNL LDRD # 10761 - Operando neutron diffraction for battery research.

## Author contributions

J.L. conceived the idea, J.L. and E.H. supervised the project. X.W. carried out the theory calculation and contributed to the analysis of corresponding results. Y. L. and S. L. collected synchrotron XRD data. A.R., S.T., Q.W. and E.H. collected and analyzed XAS data, with inputs from X. W. and J. L. Y.Z., B. S., M. L. and J.L. synthesized the material. Y.Z., A.R., Z.H. and X.R. carried out the electrochemistry test. S.Y. collected the TEM data. J.L. collected neutron diffraction data and carried out the structure analysis, with inputs from X.W. J.L., X.W. and E.H. wrote the manuscript with inputs from all authors.

## Competing interests

The authors declare no competing interests.
