## [Peer Review File · Nature Communications]

REVIEWER COMMENTS

Reviewer #1 (Remarks to the Author):

This article looks at the mechanisms at play during cycling of a P3 layered oxide as cathode for Na-ion batteries. The authors find that a new Cu-O singlet state is found during charging of the material to 4.1 V (new to the battery field, known in the cuprate superconductor field as noted by the authors). This is of potential interest, however I have some questions and concerns that may limit its interest to the broad audience of Nature Comm.

Specifically:

- the capacities are very low, this is not necessarily a deal breaker as such materials can still provide highly valuable insights; however the manner in which the capacity is low is worrisome. Specifically, the oxygen redox that is the subject of the article provides about 30 mAh/g (figure 2, 1st cycle discharge shows 30mAh/g on the 3.9 V plateau). In the SI and figure 3, it is clear that this material has nearly no Mn redox on the first cycle (this should be in the 3.0-2.0 V range). The state-of-the-art in P3 oxides to my knowledge is from Chem. Commun., 2020, 56, 10686 and J. Mater. Chem. A, 2022, 10, 251-265 (both articles are roughly the same material) where discharge capacities of 155-170 mAh/g are seen and about 70-100 mAh/g of this occurs in the 2-3 V range where the material herein shows about 20 mAh/g. So my first question is: has inducing the 30 mAh/g of singlet redox come at the loss of 70-100 mAh/g of Mn redox? I note that in the SI the Mn redox increases as the high voltage plateaus shrink with extended cycling which certainly supports my argument. It is important to recognize that the oxygen redox occurs at potentials where the electrolyte is unstable while the Mn is at lower potentials where the electrolyte is stable so this trade-off is particularly troublesome... My initial reaction is that although interesting, this singlet oxygen appears to be a state that should be avoided rather than sought after in the Mn-containing P3 materials.

-the authors refer to the oxygen redox as stable (e.g. in the title!). However, figure 2 clearly shows that the high voltage plateau where the singlet state forms reversibly shrinks with cycling (about 1/3 of this plateau is gone after only 30 cycles), even with the optimized voltage cutoff of 4.1 V. This does not strike me as stable at all. This trend should be quantified, reported and commented on as it greatly impacts the interest the field should put into researching this mechanism.

-related to this, the irreversible capacity is very high (15 mAh/g in figure 2a where the total capacity is only 75 mAh/g so 20%). The authors should comment on the source of this irreversible capacity (to my eye it appears to be primarily the 3.9 V plateau, but this should be looked at carefully).

-I think it is worth exploring why the Mn redox suppressed in this material? I think DFT could be highly informative here, it would be interesting to see what potential is predicted for the material with excess Na in it (e.g. estimate the composition if about 50 mAh/g of Mn reduction is done to insert excess Na into the pristine material).

- I do not understand what was done to obtain the Jahn-Teller distorted structures in the DFT. Were the wavefunctions projected onto distorted atomic orbitals, or was the structure actually constrained in a manner to force Mn into a J-T distorted structure? I truly don't know how to interpret the results here because of the lack of detail (I read through the methods at the end and found no details on this, if they are somewhere it might be worth repeating in the methods).

-in figure S6 (top right and bottom left panels), please zoom in on both the capacity and coulombic efficiency axes. For example, a range of 70 -110 for capacity is appropriate and 0.7-1 for CE in the top right. Including zero on the axis is highly misleading, experts might know that a CE of 95% is terrible, but the broader audience of Nat. Comm might need to be told this...

-minor edit: caption to figure S10: change Le Bai to LeBail.

In general the data is of high quality and the presentation is clear. I am not convinced this will be of interest to a broad audience and I suspect it needs to be presented as a cautionary tale to the battery community rather than a success story. I have no problem with cautionary tales, they can save me months of work!

Reviewer #2 (Remarks to the Author):

Remarks to the Author:

The paper submitted by Wang et al. employ the newly discovered P3-type $\text{Na}_{2/3}\text{Cu}_{1/3}\text{Mn}_{2/3}\text{O}_2$ as a model material to show that lattice oxygen redox can be stabilized to achieve non-hysteresis and long-term stable redox reaction. The authors experimentally (in situ XRD, ex situ soft X-ray absorption) and theoretically (DFT) confirmed that this can be achieved by forming a Zhang-Rice-like single state to provide charge compensation mechanism with stable active oxygen hole participation. The work is interesting in that it brings a concept from high temperature superconductor to sodium ion cathodes and does a deep-dive X-ray and neutron diffraction to refine the crystallographic data. This in itself is of interest to the research community. However, there are no experimental changes to the already reported material. Furthermore, it requires more work (such as structural analysis using high-resolution TEM to support the dynamic Cu-O bond length evolution

or electrochemical studies longer than 30 cycles to demonstrate stability) to convince the readership that the lattice oxygen redox reaction can be stabilized for long-term, as the presented evidence are not sufficient. The work could be published as a communication in a decent journal, but to warrant publication in Nature Communications, this work would need to be much more in-depth studies with respect to the long-term effects of the lattice oxygen redox stabilization and more novelty to providing stability to the material than just restricting the voltage window. As such, I cannot recommend that this work for publication in Nature Communications.

Comments to the authors:

Q1. The authors do an exceptional job analyzing the crystallographic structure of the P3-type $\text{Na}_{2/3}\text{Cu}_{1/3}\text{Mn}_{2/3}\text{O}_2$ material, but I am confused to how the authors are stabilizing the material other than restricting the voltage window and analyzing the redox reactions occurring at the plateaus of 3.66 V and 3.99 V.

Q2. Tying in with the stabilizing the material, the authors only cycle the material to 30 cycles. Why is this the case? To claim that the study stabilizes the P3-type $\text{Na}_{2/3}\text{Cu}_{1/3}\text{Mn}_{2/3}\text{O}_2$ material through spin singlet state, prolonged electrochemical cycling should be provided. Additionally, it would be interesting to see if the Jahn-Teller distortion is preserved at longer cycles.

Q3. The authors employed in situ X-ray absorption and ex situ X-ray absorption to experimentally show the Jahn-Teller distortion of CuO_6 octahedron preservation. It would also be interesting to verify this is indeed the case with high-resolution TEM to structurally measure the bond length.

Q4. The authors should provide all necessary information in order to reproduce DFT calculations, such as the exact version of VASP used, pseudopotentials (and their date of creation) used to model the electrons, the number of K-points and Hubbard correction used, and the combination of structural and electronic data extracted from the DFT calculation including the final energies and the coordinates of the atoms along with a zipped file containing the database.

Response to reviewer comments

Reviewer #1 (Remarks to the Author):

This article looks at the mechanisms at play during cycling of a P3 layered oxide as cathode for Na-ion batteries. The authors find that a new Cu-O singlet state is found during charging of the material to 4.1 V (new to the battery field, known in the cuprate superconductor field as noted by the authors). This is of potential interest, however I have some questions and concerns that may limit its interest to the broad audience of Nature Comm.

We truly appreciate the reviewer's positive comments. We have carefully revised our paper in light of the reviewer's suggestions. In the following, we provide point-by-point replies to the raised concerns.

Specifically:

- the capacities are very low, this is not necessarily a deal breaker as such materials can still provide highly valuable insights; however the manner in which the capacity is low is worrisome. Specifically, the oxygen redox that is the subject of the article provides about 30 mAh/g (figure 2, 1st cycle discharge shows 30mAh/g on the 3.9 V plateau). In the SI and figure 3, it is clear that this material has nearly no Mn redox on the first cycle (this should be in the 3.0-2.0 V range). The state-of-the-art in P3 oxides to my knowledge is from Chem. Commun., 2020, 56, 10686 and J. Mater. Chem. A, 2022, 10, 251-265 (both articles are roughly the same material) where discharge capacities of 155-170 mAh/g are seen and about 70-100 mAh/g of this occurs in the 2-3 V range where the material herein shows about 20 mAh/g. So my first question is: has inducing the 30 mAh/g of singlet redox come at the loss of 70-100 mAh/g of Mn redox? I note that in the SI the Mn redox increases as the high voltage plateaus shrink with extended cycling which certainly supports my argument. It is important to recognize that the oxygen redox occurs at potentials where the electrolyte is unstable while the Mn is at lower potentials where the electrolyte is stable so this trade-off is particularly troublesome... My initial reaction is that although interesting, this singlet oxygen appears to be a state that should be avoided rather than sought after in the Mn-containing P3 materials.

We thank the reviewer for the valuable comment and apologize for our negligence on comparing our results with other state-of-art Mn-based P-type sodium ion cathode materials.

Firstly, we agree that the capacity of P3-type $\text{Na}_{2/3}\text{Cu}_{1/3}\text{Mn}_{2/3}\text{O}_2$ discussed here is lower than the state-of-the-art sodium-ion layered oxide cathodes, especially compared to other Mn-based P3-type sodium ion cathodes such as the Na-Fe-Mn-O system the reviewer mentioned (*Chem. Commun.*, 2020, 56, 10686; *J. Mater. Chem. A*, 2022, 10, 251). However, it is worth mentioning that about half capacity of P3- $\text{Na}_{2/3}\text{Cu}_{1/3}\text{Mn}_{2/3}\text{O}_2$ is released at voltage higher than 3.5 V (~ 60 mAh/g), while for P3- $\text{Na}_{0.9}\text{Fe}_{0.5}\text{Mn}_{0.5}\text{O}_2$ more than half of its total capacity is released under 3.0 V (*Chem. Commun.*, 2020, 56, 10686). From the energy density point of view, the P3- $\text{Na}_{2/3}\text{Cu}_{1/3}\text{Mn}_{2/3}\text{O}_2$ material has advantages. The major difference between the current Na-Cu-Mn-O system and reported Na-Fe-Mn-O system is, Mn redox does not contribute capacity in the voltage range of 2.0 V-4.1 V for Na-Cu-Mn-O system but contributes considerable amount of capacity in this voltage range for the Na-Fe-Mn-O system. The reviewer has also mentioned this in the comments.

Secondly, to answer the question whether “inducing the 30 mAh/g of singlet redox come at the loss of 70-100 mAh/g of Mn redox?”, we need to i) establish the voltage range for Mn redox in P3-type $\text{Na}_{2/3}\text{Cu}_{1/3}\text{Mn}_{2/3}\text{O}_2$ and its capacity contribution, ii) conduct long-term cycling tests in the voltage range with/without Mn-redox and compare their redox evolution. Therefore, instead of only comparing the high-voltage behavior by setting the cycling voltage range to 2.0 V-4.1 V and 2.0 V-4.5 V as used in current manuscript, we revised the electrochemical testing condition and included three different voltage windows in the revised manuscript, namely 4.5 V-2.0 V, 4.1 V-1.5 V, and 4.1 V-2.5 V. The newly added 4.1 V-1.5 V voltage window ensures activation of Mn redox. Restraining long term cycling test to 4.1 V-2.5 V voltage window guarantees no Mn-redox participation at the beginning of cycling.

i) The electrochemical test result between 4.1 V and 1.5 V is shown in Response Fig. 1 as following. As can be seen, after the 1st charging to 4.1 V, 1st discharging to 1.5 V reveals a new plateau just below 2.0 V, corresponding to the reduction of Mn^{4+} . About 40 mAh/g capacity is released during this plateau leading to a total discharge capacity of ~130 mAh/g which is close to that of P3-type Na-rich $\text{Na}_{0.9}\text{Fe}_{0.5}\text{Mn}_{0.5}\text{O}_2$ material under a modified cycle protocol between 4.4 V and 1.5 V (*Chem. Commun.*, 2020, 56, 10686). However, in Na-Fe-Mn-O system the Mn redox occurs around 2.3 V which is higher than current Na-Cu-Mn-O system. DFT calculation predicts an even lower voltage of ~1.5 V for Mn redox activation triggered by excessive Na insertion. Qualitative agreement is reached between electrochemical measurement and DFT calculation in that Mn redox happens below 2.0 V. DFT study also revealed the reason for the low Mn redox voltage which is the increment of total energy induced by NaO_6 prisms sharing faces with TMO_6 octahedron once excessive Na is inserted into P3 structure. Since P3-type $\text{Na}_{2/3}\text{Cu}_{1/3}\text{Mn}_{2/3}\text{O}_2$ has higher content of

Mn than P3-type $\text{Na}_{0.9}\text{Fe}_{0.5}\text{Mn}_{0.5}\text{O}_2$, more NaO_6 prisms are forced into face-sharing with MnO_6 octahedron, and the total energy increases more due to large repulsion between Mn^{4+} and Na^+ . In the subsequent 2nd cycle between 1.5 V and 4.1 V, the voltage plateau corresponding to Mn redox shows good reversibility with a larger voltage hysteresis than the 3.66 V and 3.99 V plateau, which may be caused by poor Na^+ diffusivity in the Na-excessive composition.

Response Fig. 1 (a) The voltage curve of 1st charging to 4.1 V and 1st discharging to 1.5 V with blue line marking the average voltage predicted by DFT calculation for the excessive sodiation of P3-type $\text{Na}_{2/3}\text{Cu}_{1/3}\text{Mn}_{2/3}\text{O}_2$ to $\text{Na}_{5/6}\text{Cu}_{1/3}\text{Mn}_{2/3}\text{O}_2$ activating Mn redox. (b) The voltage curve of 2nd charge-discharge cycle between 1.5 V and 4.1 V, which shows good reversibility of capacity and voltage plateaus.

It is also worth comparison in the aspect of energy density between Na-Cu-Mn-O and Na-Fe-Mn-O. Despite P3- $\text{Na}_{2/3}\text{Cu}_{1/3}\text{Mn}_{2/3}\text{O}_2$ does not have very high discharge capacity, about half of it is released at voltage higher than 3.5 V (~ 60 mAh/g). In comparison, for P3- $\text{Na}_{0.9}\text{Fe}_{0.5}\text{Mn}_{0.5}\text{O}_2$, less than 1/3 of total capacity (< 40 mAh/g) is released at voltage higher than 3.5 V, and more than half of its total capacity is released under 3.0 V. This difference is not uniquely limited to the $\text{Na}_{0.9}\text{Fe}_{0.5}\text{Mn}_{0.5}\text{O}_2$ composition. For many Na-Fe-Mn-O compositions mentioned in (*J. Mater. Chem. A*, 2022, 10, 251), large portion of total capacity occurs in the voltage range under 3.5 V which is very likely related to the fact that no O redox is effectively utilized in Na-Fe-Mn-O system and low voltage Mn redox is utilized. Therefore, in respect of energy density, we believe current P3-type $\text{Na}_{2/3}\text{Cu}_{1/3}\text{Mn}_{2/3}\text{O}_2$ has non-negligible advantages.

ii) The long-term cycling performances between 4.5 V-2.0 V and 4.1 V-2.5 V are compared as shown in Response Fig. 2 and 3. In general, once the voltage window of 4.5 V-4.1V is included in electrochemical cycling, the coulombic efficiency cannot exceed 95%, charge/discharge voltage

fades significantly in limited cycles, and new redox peaks begin to appear at low voltage between 3.0 V and 2.5 V after prolonged cycles. Instead, if only the 3.66 V and 3.99 V plateaus are included in the cut-off voltage ranges such as 4.1 V-2.5 V, voltage fading is eliminated, the coulombic efficiency can be stabilized at ~ 96%, and no apparent redox peaks emerge between 3.0 V and 2.5 V. Though capacity shrink can still be observed after prolonged cycles, the capacity retention rate (63.2% after 100 cycles at moderate to slow rate) is reasonable for a non-optimized raw material.

In response to the reviewer's concerns about cycling stability and coulombic efficiency, we tried to optimize the performance through material engineering. The results indicate the synthesis temperature can be further optimized and surface coating can be introduced to improve the electrochemical performance. Through comparison between 4.5 V-2.0 V cycling and 4.1 V-2.5 V cycling, it can be clearly seen that sole inclusion of singlet redox at 3.99 V does not have much to do with Mn redox while the inclusion of higher voltage plateau at ~4.4 V may induce Mn redox activation by quickly deteriorating the material and causing degradation such as reduction of Mn^{4+} . The lost capacity due to 4.4 V plateau irreversibility is compensated by Mn redox at low voltage to some extent. Therefore, the inclusion of 4.4 V plateau comes at the expense of Mn redox. In contrast, the lost capacity after cycling between 4.1 V and 2.5 V is not accompanied by any voltage fading of 3.99 V plateau nor compensated by emergent of any low voltage peaks related to Mn redox. Therefore, the capacity loss is only related to common material aging and can be mitigated by material modification and system optimization.

Response Fig. 2 (a) Charge, discharge capacity and coulombic efficiency of the 600°C-synthesized P3-type $\text{Na}_{2/3}\text{Cu}_{1/3}\text{Mn}_{2/3}\text{O}_2$ sample between 2.0V and 4.5V (versus Na^+/Na) for 30 cycles. (b) The charge and discharge curves of cycle 1, 2, 5, 10, 20 and 30. (c) The differential voltage (dQ/dV) curves of corresponding cycles where peak intensity quickly decrease with prolonged cycles and new peaks at low voltage emerge (circled out by black line).

Response Fig. 3 (a) Charge, discharge capacity and coulombic efficiency of the 600°C-synthesized P3-type $\text{Na}_{2/3}\text{Cu}_{1/3}\text{Mn}_{2/3}\text{O}_2$ sample between 2.5 V and 4.1 V (versus Na^+/Na) for 100 cycles. (b) The charge and discharge curves of cycle 1, 2, 5, 10, 30, 50, 70, and 100. (c) The differential voltage (dQ/dV) curves of corresponding cycles where no new peaks emerge in the low voltage range (circled out by black line).

In summary, for P3-type $\text{Na}_{2/3}\text{Cu}_{1/3}\text{Mn}_{2/3}\text{O}_2$ material, **the voltage window corresponding to Mn redox is below 2.0 V, which is the reason for the absence of capacity contribution from Mn redox when cycling between 4.1 V and 2.0 V. The sole inclusion of capacity from singlet redox at 3.99 V plateau does not come at the expense of Mn redox** based on: 1) no voltage fading of 3.99 V plateau, 2) no emergence of low voltage Mn redox peak related to the 3.99 V plateau after long-term cycling. In contrast, the inclusion of capacity at high voltage plateau 4.4 V comes at the expense of Mn redox since new low voltage Mn redox peak emerges accompanying the voltage fading of 4.4 V plateau. It is worth noting that Mn redox voltage in current P3-type $\text{Na}_{2/3}\text{Cu}_{1/3}\text{Mn}_{2/3}\text{O}_2$ is lower than that measured in other representative Mn-based P-type sodium ion cathode materials. The reason for this, as revealed by DFT calculation and will-be discussed in the following, is the increment of total energy enforced by face-sharing between many NaO_6 prisms and MnO_6 octahedron when excessive Na^+ ions are inserted into P3 structure with high content of Mn.

-the authors refer to the oxygen redox as stable (e.g. in the title!). However, figure 2 clearly shows that the high voltage plateau where the singlet state forms reversibly shrinks with cycling (about 1/3 of this plateau is gone after only 30 cycles), even with the optimized voltage cutoff of 4.1 V. This does not strike me as stable at all. This trend should be quantified, reported and commented on as it greatly impacts the interest the field should put into researching this mechanism.

-related to this, the irreversible capacity is very high (15 mAh/g in figure 2a where the total capacity is only 75 mAh/g so 20%). The authors should comment on the source of this irreversible capacity (to my eye it appears to be primarily the 3.9 V plateau, but this should be looked at carefully).

Considering the above two questions are closely related, we integrated replies to them together here. We are grateful for the reviewer's comments about problems in the presentation of our work. Indeed, the number of cycles we showed in the original manuscript is not sufficient. Therefore, we re-synthesized a batch of material using the same solid-reaction method and tested its cycling performance in a much longer period with new protocol as the editor suggested (10 mA/g for 100 cycles at 25°C). The results are shown in above Response Fig. 3. Furthermore, we decomposed the capacity of each cycle to the contributions from two major plateaus as displayed in following Response Fig. 4. The capacity in the 3.75 V-3.4 V region is attributed to the 3.66 V plateau and the capacity in 4.1 V-3.75 V is attributed to the 3.99 V plateau. From these results, the capacity of two plateaus decrease at similar rate and the columbic efficiency of 3.99 V plateau is much lower than that of 3.66 V plateau. These quantifications partially agree with the reviewer's suggestion that the 3.99 V plateau of singlet redox takes the major responsibility for the overall low columbic efficiency (irreversible capacity of each cycle). Besides this, the results also demonstrate that the capacity loss along cycling is shared by two plateaus equally as the decrement rate is the same. However, it is worth noting that the capacity retention and voltage persistence (stability of the oxygen redox reaction) is still much improved compared to the oxygen redox reactions in other Na-ion cathodes with low voltage hysteresis, such as P3-type $\text{Na}_{0.6}\text{Li}_{0.2}\text{Mn}_{0.8}\text{O}_2$ (Figure 2B in *Joule* 2018, 2, 125–140) and $\text{Na}_2\text{Mn}_3\text{O}_7$ (Figure 4c in *Adv. Energy Mater.* 2018, 8, 1800409 or Figure 2b in *Chem. Mater.* 2019, 31, 10, 3756–3765). This improvement validates our argument that forming spin singlet can improve the stability of lattice oxygen redox.

We shall also point out that, despite the capacity keeps decreasing along cycling at noticeable rates, no obvious voltage fading occurs to neither plateau. There is also no new voltage plateau emerges in the low voltage region related to reduced TM cations due to material deterioration (Response Fig. 3). Therefore, it is clarified that by saying “the O redox in the form of singlet is stable”, we don't mean stability in the sense of no capacity loss or no material aging/degradation. Instead, what we try to emphasize on is the persistence of Cu-O singlet mechanism for O redox to participate in charge compensation, which causes the small and non-fading voltage hysteresis between charging and discharging. This feature, to any researcher focusing on stabilizing O redox in layer structured cathode material, is a delight to achieve.

Response Fig. 4 (a) The voltage range used to decompose the capacity contribution of each plateau marked by dotted line. (b) The charge, discharge capacity and the coulombic efficiency change during 100 cycles in the voltage range around 1st 3.66V plateau and 2nd 3.99V plateau.

As for the causes to the capacity loss and material degradation, we have some speculations such as: 1) Intrinsic defects in the synthesized material since we haven't optimized the material synthesis procedure, e.g., the sintering temperature and atmosphere etc. 2) The activated O species after 3.99 V plateau, though getting stabilized through singlet mechanism in the bulk region, is not stable near the particle surface and could be highly reactive towards organic solvents-based electrolyte or catalyze some side reactions in the surface region. These reactions related to destabilized O species near the surface would accelerate defects formation and surface deconstruction, and eventually material degradation during cycling. 3) The unoptimized electrolyte may be not compatible with the singlet mechanism stabilized O species. It has been well-recognized that the electrolyte engineering is vital for the usage of high-voltage sodium ion cathodes (*Adv. Energy Mater.* 2018, 8, 1702403; *Adv. Mater.* 2019, 31, 1808393). In current cell setting, no treatment has been done to the electrolyte. Unwanted side reactions could be severe that hurts the cycling performance. To verify these assumptions and eliminate their possible consequences, we made few attempts on the material modification. The long-term cycling performances of materials synthesized using different procedures are compared in the following Response Fig. 5.

All samples were tested under 50 mA/g for at least 100 cycles. The voltage difference between charge and discharge voltage peaks becomes larger than that in the 10 mA/g testing, which is caused by Ohm polarization due to higher current. There is still no noticeable voltage fading at this high current density. Three tested samples are the ones synthesized under 600°C, synthesized under 675°C, and synthesized under 675°C surface-coated with trace amounts of Al-Mg-Ti dopants.

Increasing the synthesis temperature from 600°C to 675°C can increase the overall charge capacity by about 5 mAh/g throughout the cycling. At 675°C, introducing Al-Mg-Ti surface coating can systematically improve the cycling performance. Compared to the 600°C sample, overall charge/discharge capacity increased by about 10 mAh/g, 1st cycle CE increases from 81.58% to 87.03%, and the discharge capacity retention after 100 cycles increases from 70.82% to 73.45%. For the 675°C sample with Al-Mg-Ti coating, the voltage peak corresponding oxygen singlet redox even gets sharper after 100 cycles which highlights the effectiveness of material optimization in enhancing the columbic efficiency of 3.99 V plateau.

Response Fig. 5 The 50 mA/g cycling performances, representative voltage profiles, and corresponding differential voltage curves of P3-type $\text{Na}_{2/3}\text{Cu}_{1/3}\text{Mn}_{2/3}\text{O}_2$ samples synthesized at different conditions: (a-c) a regular 600°C solid-state synthesis; (d-f) solid-state synthesis at elevated temperature of 675°C; (g-i) solid-state synthesis at 675°C with multi-element trace doping of Al, Mg, and Ti.

Through showing the effects of material optimization on the cycling performance, we demonstrated the huge potential of the new singlet mechanism for O redox utilization. In addition, we strongly believe that electrolyte engineering can further enhance the cycling performance of P3-type $\text{Na}_{2/3}\text{Cu}_{1/3}\text{Mn}_{2/3}\text{O}_2$. However, it deviates from the focus of current work on reporting the new mechanism in cathode material. Therefore, the effects of electrolyte optimization have not been explored yet. The current situation of P3-type $\text{Na}_{2/3}\text{Cu}_{1/3}\text{Mn}_{2/3}\text{O}_2$ is similar to that of Ni-rich material at early stages of development. They also suffer from relatively low columbic efficiency and instability during long-term high voltage cycling (*Adv. Funct. Mater.*, 2020, 30, 2004748; *Chem. Soc. Rev.*, 2020, 49, 4667). However, after years of optimization, many effective strategies have been used to stabilize the material which helps paving the road for practical applications. One last thing we want to point out is that no matter how we altered the material synthesis procedure, the 3.66 V and 3.99 V plateaus remain relatively stable, and the feature of small voltage hysteresis persists, which once again demonstrates the perseverance of Cu-O singlet O redox mechanism.

-I think it is worth exploring why the Mn redox suppressed in this material? I think DFT could be highly informative here, it would be interesting to see what potential is predicted for the material with excess Na in it (e.g. estimate the composition if about 50 mAh/g of Mn reduction is done to insert excess Na into the pristine material).

We thank the reviewer for the valuable suggestions. Follow the guidance, we adopted DFT calculations to explore the theoretical potential plateaus corresponding to Mn redox. By introducing excessive Na atoms into the model of P3-type $\text{Na}_{2/3}\text{Cu}_{1/3}\text{Mn}_{2/3}\text{O}_2$, we built the P3 $\text{Na}_{5/6}\text{Cu}_{1/3}\text{Mn}_{2/3}\text{O}_2$ model as shown in the following Response Fig. 6. The in-plane Na-Vacancy configuration of ground state was approximated using the same method for $\text{Na}_{1/2}\text{Cu}_{1/3}\text{Mn}_{2/3}\text{O}_2$ and $\text{Na}_{1/6}\text{Cu}_{1/3}\text{Mn}_{2/3}\text{O}_2$ models.

The theoretical average potential for Mn redox with excessive sodiation is then predicted by Nernst Equation with DFT calculated total energies. The prediction is around 1.5 V vs. Na^+/Na as shown in Response Fig. 1. Qualitative agreement is reached between DFT prediction and electrochemical measurement in that Mn-redox occurs just below 2.0 V. Quantitative difference exists due to limited cell size and insufficient Na-Vacancy configuration sampling in DFT calculation. Though the number does not match exactly with the experimental voltage curve, insight on the reason for low Mn redox potential can still be obtained. Clearly, due to the layer arrangements in P3 structure, excessive sodiation would result in face-sharing between NaO_6 prisms and TMO_6 octahedron, leading to a different scenario from the pristine state where they only share edges. Face sharing

would shorten the distance between Na^+ and TM cations leading to large repulsion between them and increasing the total energy of P3 materials. Mn^{4+} - Na^+ repulsion is strong due to high valence state of Mn^{4+} . Therefore, higher amount of Mn would cause higher total energy of excessively sodiated P3 material which lowers the energy release during sodiation and the corresponding sodiation potential. P3-type $\text{Na}_{2/3}\text{Cu}_{1/3}\text{Mn}_{2/3}\text{O}_2$ materials has higher content of Mn^{4+} cations than most Na-Fe-Mn-O materials (*Chem. Commun.*, 2020, 56, 10686; *J. Mater. Chem. A*, 2022, 10, 251), leading to a much lower Mn redox potential. The “suppression” of Mn redox is mostly related to the layer arrangement in P3 structure and the increment of NaO_6 site energies due to high Mn content but is unlikely related to the Cu-O singlet redox.

Response Fig. 6 The illustrative showing of structural model used by DFT calculation for excessively sodiated state $\text{Na}_{5/6}\text{Cu}_{1/3}\text{Mn}_{2/3}\text{O}_2$ in (a-b) side view, top view of (c) Cu-Mn layer and top view of (d) Na layer with Na^+ ions occupying the prismatic sites and sharing faces with TM cations.

- I do not understand what was done to obtain the Jahn-Teller distorted structures in the DFT. Were the wavefunctions projected onto distorted atomic orbitals, or was the structure actually constrained in a manner to force Mn into a J-T distorted structure? I truly don't know how to interpret the results

here because of the lack of detail (I read through the methods at the end and found no details on this, if they are somewhere it might be worth repeating in the methods).

We apologize for missing the description about the construction of models with/without CuO_6 JT distortion. We obtained the structural model without CuO_6 JT distortion through regular DFT structural relaxation with GGA+U type functional. Since DFT+U cannot address the Cu-O singlet state with strong electron correlation, the relaxed structures contain highly symmetrical CuO_6 octahedra without JT distortion. Then based on the DFT+U relaxed cell size and shape, we constructed the model with CuO_6 JT distortion by manually setting the coordination of TM cations and O anions to those refined from in-situ XRD results. The TM and O atom position and cell size/shape were kept fixed during following relaxation except that the position of Na^+ was allowed to move. Then electronic structures of the model with and without CuO_6 JT distortion were calculated using the same HSE06 type hybrid functional for us to obtain accurate description of empty states at bottom of valence band. Those descriptions have now been updated into the methods part of revised manuscript.

-in figure S6 (top right and bottom left panels), please zoom in on both the capacity and coulombic efficiency axes. For example, a range of 70 -110 for capacity is appropriate and 0.7-1 for CE in the top right. Including zero on the axis is highly misleading, experts might know that a CE of 95% is terrible, but the broader audience of Nat. Comm might need to be told this...

We thank the reviewer for the suggestions. The changes have been made to figures in the revised manuscript accordingly.

-minor edit: caption to figure S10: change Le Bai to LeBail.

Corrections have been made accordingly.

In general the data is of high quality and the presentation is clear. I am not convinced this will be of interest to a broad audience and I suspect it needs to be presented as a cautionary tale to the battery community rather than a success story. I have no problem with cautionary tales, they can save me months of work!

We thank the reviewer for the high appraisal of our work and pointing out our problem in presenting the work. With the replies to previous questions, we hope we have convinced the reviewer that the newly found Cu-O singlet mechanism for non-hysteresis O redox utilization has huge potential for future applications. We believe these findings shall be presented as a success in utilizing lattice

O redox reaction in sodium ion cathode material, and it is worth further investigation instead of a sugar-coated devil that needs to be avoided. We make this statement based on following points:

1) Though the capacity release of P3-type $\text{Na}_{2/3}\text{Cu}_{1/3}\text{Mn}_{2/3}\text{O}_2$ material is not as high as that of Na-Fe-Mn-O systems reported by literatures in the voltage range of 4.4 V-2.0 V (*Chem. Commun.*, 2020, 56, 10686; *J. Mater. Chem. A*, 2022, 10, 251), more than half of the capacity is release in the high voltage range above 3.5 V thanks to usage of O redox instead of the utilization of Mn redox below 3.0 V. In fact, if cycling voltage is restricted to 4.1 V-2.5 V, capacity release of current P3 $\text{Na}_{2/3}\text{Cu}_{1/3}\text{Mn}_{2/3}\text{O}_2$ material is almost the same as that of P3 $\text{Na}_{0.9}\text{Fe}_{0.5}\text{Mn}_{0.5}\text{O}_2$ (*Chem. Commun.*, 2020, 56, 10686) but the average voltage is much higher. In the respect of energy density, P3 $\text{Na}_{2/3}\text{Cu}_{1/3}\text{Mn}_{2/3}\text{O}_2$ has great advantages.

2) Though the long-term cycling performance of untreated raw material is not impressive at the moment, the performance can be drastically improved if further optimization (e.g., synthetic conditions, surface coating and electrolyte modifications etc.) is introduced. This study suggests the intrinsic property of bulk material is promising and further material/system engineering is needed to unlock its full potential.

3) Most importantly, successful utilization of O redox with nearly zero hysteresis has always been a grand challenge for anion redox in cathode materials. We showcased in this report that over 100 cycles can be realized in P3 $\text{Na}_{2/3}\text{Cu}_{1/3}\text{Mn}_{2/3}\text{O}_2$. The Cu-O singlet electronic state proves itself as a persistent mechanism for O redox stabilization. For comparison, cycling performances of the other small voltage hysteresis anion redox cathodes, such as $\text{Na}_2\text{Mn}_3\text{O}_7$ and P3-type $\text{Na}_{0.6}\text{Li}_{0.2}\text{Mn}_{0.8}\text{O}_2$ are often limited to 20-30 cycles (*Joule* 2018, 2, 125–140; *Adv. Energy Mater.* 2018, 8, 1800409; *Chem. Mater.* 2019, 31, 10, 3756–3765).

Reviewer #2 (Remarks to the Author):

The paper submitted by Wang et al. employ the newly discovered P3-type $\text{Na}_{2/3}\text{Cu}_{1/3}\text{Mn}_{2/3}\text{O}_2$ as a model material to show that lattice oxygen redox can be stabilized to achieve non-hysteresis and long-term stable redox reaction. The authors experimentally (in situ XRD, ex situ soft X-ray absorption) and theoretically (DFT) confirmed that this can be achieved by forming a Zhang-Rice-like single state to provide charge compensation mechanism with stable active oxygen hole participation. The work is interesting in that it brings a concept from high temperature superconductor to sodium ion cathodes and does a deep-dive X-ray and neutron diffraction to refine the crystallographic data. This in itself is of interest to the research community. However, there are no experimental changes to the already reported material. Furthermore, it requires more work (such as structural analysis using high-resolution TEM to support the dynamic Cu-O bond length evolution or electrochemical studies longer than 30 cycles to demonstrate stability) to convince the readership that the lattice oxygen redox reaction can be stabilized for long-term, as the presented evidence are not sufficient. The work could be published as a communication in a decent journal, but to warrant publication in Nature Communications, this work would need to be much more in-depth studies with respect to the long-term effects of the lattice oxygen redox stabilization and more novelty to providing stability to the material than just restricting the voltage window. As such, I cannot recommend that this work for publication in Nature Communications.

We are truly grateful for the reviewer's comments and appraisal of our work's novelty. Enlightened by the reviewer's suggestions, we added the long cycling performance, comparison of electrochemical test behavior in different voltage ranges, TEM characterization of local structures, more detailed discussion about comparison with other established high-capacity P-type sodium-ion cathodes, and DFT calculation details. In the following point-to-point replies, we hope to highlight the importance of our work (particularly on the importance of using spin singlet to stabilize lattice oxygen redox reactions) and stimulate potential interests to the broad reader community of *Nature Communications*.

Comments to the authors:

Q1. The authors do an exceptional job analyzing the crystallographic structure of the P3-type $\text{Na}_{2/3}\text{Cu}_{1/3}\text{Mn}_{2/3}\text{O}_2$ material, but I am confused to how the authors are stabilizing the material other than restricting the voltage window and analyzing the redox reactions occurring at the plateaus of 3.66 V and 3.99 V.

Q2. Tying in with the stabilizing the material, the authors only cycle the material to 30 cycles. Why is this the case? To claim that the study stabilizes the P3-type $\text{Na}_{2/3}\text{Cu}_{1/3}\text{Mn}_{2/3}\text{O}_2$ material through spin singlet state, prolonged electrochemical cycling should be provided. Additionally, it would be interesting to see if the Jahn-Teller distortion is preserved at longer cycles.

Reply: Since Q1 and Q2 are tied closely together, we integrated answers to them here.

We truly appreciate the reviewer for pointing out the deficiency in our original manuscript. Indeed, the cycling performance we showed in the current manuscript is far from ideal. However, it is important to note that the cycling performance of the current material is the best among various cathode materials using lattice oxygen redox with small voltage hysteresis, such as $\text{Na}_2\text{Mn}_3\text{O}_7$ and P3-type $\text{Na}_{0.6}\text{Li}_{0.2}\text{Mn}_{0.8}\text{O}_2$ (*Joule* 2018, 2, 125–140; *Adv. Energy Mater.* 2018, 8, 1800409; *Chem. Mater.* 2019, 31, 10, 3756–3765). Successfully utilizing lattice oxygen redox with non or small voltage hysteresis has always been a grand challenge for using anion redox in cathode materials. It showcased in this report that over 100 cycles can be realized in P3 $\text{Na}_{2/3}\text{Cu}_{1/3}\text{Mn}_{2/3}\text{O}_2$ with reasonable capacity retention. The Cu-O singlet electronic state proves itself as a persistent mechanism for O redox stabilization. Simply restricting the voltage window during cycling cannot explore all the possible stabilizing strategies. To explore plausible ways to improve cyclability at the material level other than electrochemical testing protocol, we made efforts in the following aspects: **i)** Comparing the electrochemical behavior of the material in different voltage ranges which includes the activation of Mn redox and pushing to wider voltage cut-off windows. **ii)** Tuning the synthetic conditions for materials and comparing their long-term cycling performances within the optimized voltage window. These efforts help analyzing potential degradation mechanism and possible stabilizing strategies.

Three different voltages cut-off windows are included in the revised manuscript now, namely 4.1 V-1.5 V, 4.5 V-2.0 V, and 4.1 V-2.5 V. The newly added 4.1 V-1.5 V voltage window ensures activation of Mn redox. In contrast, restricting long term cycling test to 4.1 V-2.5 V guarantees the sole charge compensation contribution from interested redox during the cycling.

i) The electrochemical test result between 4.1 V and 1.5 V is shown in Response Fig. 1 as following. After the 1st charging to 4.1 V, 1st discharging to 1.5 V reveals a new plateau just below 2.0 V, which corresponds to the reduction of Mn^{4+} . About 40 mAh/g capacity is released during this plateau, leading to a total discharge capacity of ~130 mAh/g which is close to that of P3-type Na-rich $\text{Na}_{0.9}\text{Fe}_{0.5}\text{Mn}_{0.5}\text{O}_2$ material under a modified cycle protocol between 4.4 V and 1.5 V (*Chem. Commun.*, 2020, 56, 10686). However, in Na-Fe-Mn-O system the Mn redox occurs around 2.3 V

which is higher than current Na-Cu-Mn-O system. DFT calculation predicts an even lower voltage of ~ 1.5 V for Mn redox activation triggered by excessive Na insertion. Qualitative agreement is reached between electrochemical measurement and DFT calculation in that Mn redox happens below 2.0 V. DFT study also revealed that the reason for the relatively low Mn redox potential is the increment of total energy induced by the face-sharing between NaO_6 prisms and TMO_6 octahedron once excessive Na^+ are inserted. Since P3-type $\text{Na}_{2/3}\text{Cu}_{1/3}\text{Mn}_{2/3}\text{O}_2$ has higher content of Mn than P3-type $\text{Na}_{0.9}\text{Fe}_{0.5}\text{Mn}_{0.5}\text{O}_2$, more NaO_6 prisms are forced into face-sharing with MnO_6 octahedron, and the total energy increases more due to large repulsion between Mn^{4+} and Na^+ . In the subsequent 2nd cycle between 1.5 V and 4.1 V, the voltage plateau corresponding to Mn redox shows good reversibility but with a larger voltage hysteresis than those for the 3.66 V and 3.99 V plateaus. This may be caused by poor Na^+ diffusivity in the Na-excessive composition. Due to the relatively low redox potential of Mn redox, we did not adopt this voltage range for further long-term cycling.

Response Fig. 1 (a) The voltage curve of 1st charging to 4.1 V and 1st discharging to 1.5 V with blue line marking the average voltage predicted by DFT calculation for the excessive sodiation of P3-type $\text{Na}_{2/3}\text{Cu}_{1/3}\text{Mn}_{2/3}\text{O}_2$ to $\text{Na}_{5/6}\text{Cu}_{1/3}\text{Mn}_{2/3}\text{O}_2$ activating Mn redox. (b) The voltage curve of 2nd charge-discharge cycle between 1.5 V and 4.1 V, which shows good reversibility of capacity and voltage plateaus.

The electrochemical performances between 4.5 V-2.0 V and 4.1 V-2.5 V are compared in longer term cycling as shown in following Response Fig. 2 and 3. In general, once the voltage range 4.5 V-4.1V is included in electrochemical cycling, the coulombic efficiency cannot exceed 95%, charge/discharge voltage fades significantly in limited cycles, and new redox peaks begin to appear at low voltage between 3.0 V and 2.5 V after prolonged cycles. Instead, if only the 3.66 V and 3.99 V plateaus are included in the cut-off voltage ranges such as 4.1 V-2.5 V, voltage fading is

eliminated, the coulombic efficiency can be stabilized at $\sim 96\%$, and no apparent redox peaks emerge between 3.0 V and 2.5 V. Though the capacity fading can still be observed after prolonged cycles, the capacity retention rate (63.2% after 100 cycles) is reasonable for a non-optimized raw material, particular for those utilizing lattice oxygen redox reactions. Through the comparison between the cycling performance with different cut-off windows, i.e., 4.5 V-2.0 V and 4.1 V-2.5 V, it can be seen that the inclusion of higher voltage plateau at ~ 4.4 V may induce Mn redox activation by quickly deteriorating the material and causing degradation such as reduction of Mn^{4+} . The capacity loss due to the irreversibility of the 4.4 V plateau is partially compensated by the Mn redox at low voltage to some extent. In contrast, the capacity loss after cycling between 4.1 V and 2.5 V is not accompanied by any voltage fading of the 3.99 V plateau nor compensated by emergent of any low voltage peaks related to the Mn redox. Therefore, the capacity loss is only related to common material aging or surface parasitic reactions and may be effectively mitigated by surface modification and synthetic condition optimization.

Response Fig. 2 (a) Charge, discharge capacity and coulombic efficiency of the 600°C -synthesized P3-type $\text{Na}_{2/3}\text{Cu}_{1/3}\text{Mn}_{2/3}\text{O}_2$ sample between 2.0V and 4.5V (versus Na^+/Na) for 30 cycles. (b) The charge and discharge curves of cycle 1, 2, 5, 10, 20 and 30. (c) The differential voltage (dQ/dV) curves of corresponding cycles where peak intensity quickly decrease with prolonged cycles and new peaks at low voltage-4.5V (circled out by black line).

Response Fig. 3 (a) Charge, discharge capacity and coulombic efficiency of the 600°C -synthesized P3-type $\text{Na}_{2/3}\text{Cu}_{1/3}\text{Mn}_{2/3}\text{O}_2$ sample between 2.5 V and 4.1 V (versus Na^+/Na) for 100 cycles. (b) The

charge and discharge curves of cycle 1, 2, 5, 10, 30, 50, 70, and 100. (c) The differential voltage (dQ/dV) curves of corresponding cycles where no new peaks emerge in the low voltage range (circled out by black line).

ii) We re-synthesized a batch of material using the same solid-reaction method and tested its cycling performance between 4.1 V and 2.5 V in a much longer period with new protocol as the editor suggested (10 mA/g for 100 cycles at 25°C). The results are shown in above Response Fig. 3. Furthermore, we decomposed the capacity of each cycle to the contributions from two major plateaus as displayed in following Response Fig. 4. The capacity in the 3.75 V-3.4 V region is attributed to the 3.66 V plateau and the capacity in 4.1 V-3.75 V is attributed to the 3.99 V plateau. From the results, it can be seen that the capacity of two plateaus decrease at similar rate and the columbic efficiency of 3.99 V plateau is much lower that of 3.66 V plateau. These quantifications suggest that the 3.99 V plateau makes the major contribution to the overall low columbic efficiency (irreversible capacity of each cycle). Beside this, the results also demonstrate that the capacity loss along cycling is shared by two plateaus equally as the decrement rate is the same.

Response Fig. 4 (a) The voltage range used to decompose the capacity contribution of each plateau marked by dotted line. (b) The charge, discharge capacity and the coulombic efficiency change during 100 cycles in the voltage range around 1st 3.66V plateau and 2nd 3.99V plateau.

We must point out that, though the capacity keeps decreasing along cycling, no obvious voltage fading occurs for neither plateau nor new plateau emerges in the low voltage region related to reduced TM cations due to material degradation (Response Fig. 3). Therefore, the Cu-O singlet mechanism is persistent for lattice oxygen redox to participate in charge compensation, which leads to the small and non-fading voltage hysteresis between charging and discharging. This feature, to

any researcher focusing on stabilizing lattice oxygen redox in layer structured cathode material, is a delight to achieve.

As for the causes to the capacity loss and material degradation, we have some speculations such as:

- 1) Intrinsic defects in the synthesized material since no optimization on the synthetic procedures.
- 2) The activated oxidized species on particle surface after 3.99 V plateau. Although singlet mechanism can stabilize lattice oxygen redox in the bulk region, is not stable near the surface of material particle and could be highly reactive towards electrolyte components or catalyze side reactions on the particle surface. These reactions related to destabilized oxidized oxygen species near the surface would accelerate defects formation and surface deconstruction, and eventually material degradation during cycling.
- 3) The unoptimized electrolyte may be not compatible with the singlet mechanism stabilized O species. It has been well-recognized that the electrolyte engineering is vital for the usage of high-voltage sodium ion cathodes (*Adv. Energy Mater.* 2018, 8, 1702403; *Adv. Mater.* 2019, 31, 1808393). In current cell setting, no treatment has been done to the electrolyte. Unwanted side reactions could be severe that deteriorate the cycling performance. To verify these assumptions and eliminate their possible consequences, we made few attempts on the cathode material modification. The long-term cycling performances of materials synthesized using different procedures are compared in the following Response Fig. 5.

All samples were tested under 50 mA/g for at least 100 cycles. The voltage difference between charge and discharge voltage peaks becomes larger than that in 10 mA/g testing, which is caused by Ohm polarization due to the use of higher current density. There is still no noticeable voltage fading of any kind. Three tested samples are the ones synthesized under 600°C, synthesized under 675°C, and synthesized under 675°C with trace doping of Al-Mg-Ti dopants. Increasing the synthesis temperature from 600°C to 675°C can increase the overall charge capacity by about 5 mAh/g throughout the cycling. At 675°C, introducing Al-Mg-Ti coating can systematically improve the cycling performance. Compared to the 600°C sample, overall charge/discharge capacity increases by about 10 mAh/g, 1st cycle CE increases from 81.58% to 87.03%, and the discharge capacity retention after 100 cycles increases from 70.82% to 73.45%. For the 675°C sample with Al-Mg-Ti coating, the voltage peak corresponding O singlet redox even gets sharper after 100 cycles which highlights the effectiveness of material optimization in enhancing the columbic efficiency of 3.99 V plateau.

Response Fig. 5 The 50 mA/g cycling performances, representative voltage profiles, and corresponding differential voltage curves of P3-type $\text{Na}_{2/3}\text{Cu}_{1/3}\text{Mn}_{2/3}\text{O}_2$ samples synthesized at different conditions: (a-c) a regular 600°C solid-state synthesis; (d-f) solid-state synthesis at elevated temperature of 675°C; (g-i) solid-state synthesis at 675°C with multi-element trace doping of Al, Mg, and Ti.

It is worth noting that regardless of the material synthetic conditions, the 3.66 V and 3.99 V plateaus remains relatively flat, and the feature of small voltage hysteresis persists, which once again demonstrates the perseverance of Cu-O singlet O redox mechanism. Through showing the effects of material optimization on the cycling performance, we demonstrated that intrinsic bulk property of P3-type $\text{Na}_{2/3}\text{Cu}_{1/3}\text{Mn}_{2/3}\text{O}_2$ is very promising. We believe electrolyte engineering can further enhance the cycling performance of this compound. However, it deviates from the focus of current work on reporting the new mechanism in this type of cathode material. Therefore, the effects of electrolyte optimization have not been explored yet. The current situation of P3-type $\text{Na}_{2/3}\text{Cu}_{1/3}\text{Mn}_{2/3}\text{O}_2$ is actually similar to that of Ni-rich material at early stages of development. They

also suffer from low coulombic efficiency and instability during long-term high voltage cycling (*Adv. Funct. Mater.*, 2020, 30, 2004748; *Chem. Soc. Rev.*, 2020, 49, 4667). However, after years of optimization, many effective strategies have been developed to stabilize the material which help them find wide market in practical applications.

As the reviewer pointed out, JT distortion preservation after long-term cycling is important to prove the stability of the newly discovered ZR-singlet mechanism for O redox. We, in the light of reviewer's guidance, conducted EXAFS test of samples after 100 cycles at 10 mA/g and 238 cycles at 50 mA/g and compared them with one after 1st cycle. Results are shown in following Response Fig. 6. The side peak around 2Å in EXAFS curve marks the existence of CuO₆ JT distortion as analyzed in the supporting information to manuscript. In Response Fig.6, it can be clearly seen that this side peak is preserved after long-term cycling at different rates. The CuO₆ JT distortion is persistent after cycling which suggests the stability of Cu-O singlet mechanism for O redox activation.

Response Fig. 6 Comparison between Fourier transformed EXAFS spectra in real space at Cu K-edge of samples after different number of cycles. Dotted grey line marks the side peak

corresponding to the short Cu-O bond in JT-distorted CuO_6 octahedron which suggests the preservation of JT distortion in P3-type $\text{Na}_{2/3}\text{Cu}_{1/3}\text{Mn}_{2/3}\text{O}_2$ after long-term cycling.

Q3. The authors employed in situ X-ray absorption and ex situ X-ray absorption to experimentally show the Jahn-Teller distortion of CuO_6 octahedron preservation. It would also be interesting to verify this is indeed the case with high-resolution TEM to structurally measure the bond length.

We thank the reviewer for the valuable suggestions. In the revised manuscript, we employed the high-resolution TEM to characterize the particle of P3 $\text{Na}_{2/3}\text{Cu}_{1/3}\text{Mn}_{2/3}\text{O}_2$ sample. Results are shown in following Response Fig. 7. The morphological observation shows that the particle is about 1~2 μm in diameter and has perfect layer structure. Unfortunately, since the MnO_6 octahedra and CuO_6 octahedra stack together, HR-TEM fails to distinguish the long and short Cu-O bond due to JT distortion. In the samples after long-term cycling, the quality of TEM observation can only be too vague to enable differentiation of long and short Cu-O bonds. Maybe HAADF-STEM or AI-assisted super resolution TEM techniques can monitor CuO_6 distortion but will still be very challenge due to the presence of stacking faults. Indeed, the JT distortion of CuO_6 have been observed in both neutron PDF and Cu K-edge EXAFS, two robust short range characterization tools that have been broadly used to characterize orbital distortions. We updated the TEM results into the revised manuscript and use the EXAFS results after long-term cycling to support the stability of CuO_6 JT distortion.

Response Fig. 7 (a) Low magnification TEM images showing the morphology of the synthesized P3-type $\text{Na}_{2/3}\text{Cu}_{1/3}\text{Mn}_{2/3}\text{O}_2$ material particles. (b) Electron diffraction pattern of selected area along the [013] direction with the marked spots corresponding to the (100) and (031) diffraction plane. (c) High resolution TEM image showing the cross section of (100) lattice planes in the P3-type $\text{Na}_{2/3}\text{Cu}_{1/3}\text{Mn}_{2/3}\text{O}_2$ particle which demonstrates the perfect layered structure of synthesized samples.

Q4. The authors should provide all necessary information in order to reproduce DFT calculations, such as the exact version of VASP used, pseudopotentials (and their date of creation) used to model the electrons, the number of K-points and Hubbard correction used, and the combination of structural and electronic data extracted from the DFT calculation including the final energies and the coordinates of the atoms along with a zipped file containing the database.

We thank the reviewer for the valuable suggestion. All details have been integrated to the methods part of the revised manuscript. The input files and key output files containing the requested information for all calculations are now zipped up and provided as database file to the revised manuscript.

In light of the comments from reviewers, we have thoroughly modified the manuscript in following aspects:

1, we changed the discussion part on long term cycling performance: 1) the capacity is lower but only because our voltage range is narrower and higher, which leads to a higher effective energy density. 2) we clarified Mn-redox voltage is below 2V different from the Fe-Mn system or other P-type sodium-ion cathode material. We attributed this difference to the forced NaO_6 - TMO_6 face sharing at high sodiation state due to the P3 structure. 3) We demonstrated that the rate-capability and long-cycling performance can be enhanced by synthesis procedure optimization: higher synthesis temperature and elemental doping, for example. Better crystallinity of sample and protected particle surface is beneficial. Further electrolyte optimization and interphase engineering may be critical to unleash the full potential of Cu-Mn sodium-ion cathode material.

2, On the preservation of JT distortion after cycling, we added the discussion to the EXAFS part pointing to the measured EXAFS data in SI after long-term cycling. We also mentioned that HR-TEM fails to clearly measure the different Cu-O bond length because CuO_6 and MnO_6 stacks together and cannot be distinguished from each other.

3, We modified the discussion on DFT results, especially the significant change of BoC states once the Na content is smaller than $\text{Na}_{1/2}$ entering the 3.99V plateau. We expanded the DFT calculation part to include the model structure building, enumeration method used to generate Na-Vac configuration, and JT distortion introduction.

REVIEWERS' COMMENTS

Reviewer #1 (Remarks to the Author):

The authors did a very good job of addressing my main concerns, particularly with understanding what happened to the Mn redox.

I however urge them to quantify energy density and fix the statement:

"the long and highly reversible 3.66 V and 3.99 V plateaus give current material 184 advantages in the aspect of energy density"

This is in comparing their P3 to the state-of-the-art. When I look at the state-of-the-art that the authors state I get energy densities of about 380-400 Wh/kg, while for this article I get closer to 300 Wh/kg. I don't think this material is competitive on energy density, even with the anionic redox and this article should not be accepted with an error of this type. Otherwise, I feel it is ready to accept.

Reviewer #2 (Remarks to the Author):

The authors address the concerns of the reviewer with detailed responses and have improved the quality of the manuscript. The reviewer recommends this manuscript for publication in Nature Communications.

Response to reviewer comments

Response to Reviewer Comments:

Reviewer #1 (Remarks to the Author):

The authors did a very good job of addressing my main concerns, particularly with understanding what happened to the Mn redox.

We thank the reviewer for going through the revised manuscript and truly appreciate the reviewer's affirmation of our efforts.

I however urge them to quantify energy density and fix the statement: "the long and highly reversible 3.66 V and 3.99 V plateaus give current material 184 advantages in the aspect of energy density" This is in comparing their P3 to the state-of-the-art. When I look at the state-of-the-art that the authors state I get energy densities of about 380-400 Wh/kg, while for this article I get closer to 300 Wh/kg. I don't think this material is competitive on energy density, even with the anionic redox and this article should not be accepted with an error of this type. Otherwise, I feel it is ready to accept.

We thank the reviewer for pointing out the misinterpretation of electrochemical data in our revision. To quantify the energy density of current reported P3-type $\text{Na}_{2/3}\text{Cu}_{1/3}\text{Mn}_{2/3}\text{O}_2$, we listed the change of discharge energy density upon cycling by integrating the voltage against discharge capacity (Table R1). For the undoped sample synthesized at 600°C, when cycled at 10mA/g the 1st discharge capacity is ~79.80 mAh/g and the corresponding discharge energy density is ~288.28 Wh/kg. From Table R1, it's clear that current energy density cannot compete with that of state-of-art high energy density sodium ion cathode materials such as P3- $\text{Na}_{0.9}\text{Fe}_{0.5}\text{Mn}_{0.5}\text{O}_2$ (~430Wh/kg), which has been pointed out by the reviewer. Therefore, we modified our description in the revised manuscript.

However, it is worth noting that current energy density reaches about 2/3 that of P3- $\text{Na}_{0.9}\text{Fe}_{0.5}\text{Mn}_{0.5}\text{O}_2$ while the capacity is only about half. Therefore, the long plateau at high voltage of 3.66V and 3.99V are still beneficial for improving the energy density of current material. Furthermore, upon cycling the energy density retention rate is almost equivalent to the capacity retention rate for current P3-type $\text{Na}_{2/3}\text{Cu}_{1/3}\text{Mn}_{2/3}\text{O}_2$ material, which indicates the stability of high voltage plateau. Therefore, the focus of further optimization should be improving capacity release

and retention. The long and stable high voltage plateau of P3-type $\text{Na}_{2/3}\text{Cu}_{1/3}\text{Mn}_{2/3}\text{O}_2$ material is still an advantage in respect of electrochemical performance.

Cycle	Discharge Capacity (mAh/g)	Retention	Discharge Energy Density (Wh/kg)	Retention
1	79.80	--	288.28	--
Ref	155	51.48%	430	67.04%
2	79.73	99.91%	287.36	99.68%
5	78.74	98.67%	283.04	98.18%
10	76.61	96.00%	274.52	95.23%
30	69.76	87.42%	248.72	86.28%
50	63.95	80.14%	226.41	78.49%
70	58.36	73.13%	205.12	71.15%
100	50.40	63.16%	175.17	60.76%

Ref: “Introducing Na-sufficient P3- $\text{Na}_{0.9}\text{Fe}_{0.5}\text{Mn}_{0.5}\text{O}_2$ as a cathode material for Na-ion batteries”, *Chem. Commun.*, **2020**, 56, 10686

In the newly revised manuscript, we have modified the statement mentioned by the reviewer and the revision is as following:

“This difference of Mn redox occurring at lower voltage leads to a lower capacity in the voltage range 4.1 V-2.0 V compared to other reported state-of-art P3 type materials. “For example, the initial discharge capacity of the current P3- $\text{Na}_{2/3}\text{Cu}_{1/3}\text{Mn}_{2/3}\text{O}_2$ material (between 4.1V and 2V) is only about half the capacity of that of P3- $\text{Na}_{0.9}\text{Fe}_{0.5}\text{Mn}_{0.5}\text{O}_2$ between 4.4V and 1.5V (79.8 mAh/g vs. 155 mAh/g).⁵³ However, the initial discharge energy density of the P3- $\text{Na}_{2/3}\text{Cu}_{1/3}\text{Mn}_{2/3}\text{O}_2$ reaches 67% of that in the P3- $\text{Na}_{0.9}\text{Fe}_{0.5}\text{Mn}_{0.5}\text{O}_2$ (288.28 Wh/kg vs. 430 Wh/kg), due to the larger capacity and higher voltage associated with the 3.66 V and 3.99 V plateaus.”

Reviewer #2 (Remarks to the Author):

The authors address the concerns of the reviewer with detailed responses and have improved the quality of the manuscript. The reviewer recommends this manuscript for publication in Nature Communications.

We are truly grateful for this positive feedback and thank the reviewer for appraising the modified manuscript.